# Neural Signed Distance Function Inference through Splatting 3D Gaussians Pulled on Zero-Level Set

**Wenyuan Zhang**[1], **Yu-Shen Liu**[1][*], **Zhizhong Han**[2]
School of Software, Tsinghua University, Beijing, China[1]
zhangwen21@mails.tsinghua.edu.cn, liuyushen@tsinghua.edu.cn
Department of Computer Science, Wayne State University, Detroit, USA[2]
h312h@wayne.edu

## Abstract

It is vital to infer a signed distance function (SDF) in multi-view based surface reconstruction. 3D Gaussian splatting (3DGS) provides a novel perspective for volume rendering, and shows advantages in rendering efficiency and quality. Although 3DGS provides a promising neural rendering option, it is still hard to infer SDFs for surface reconstruction with 3DGS due to the discreteness, the sparseness, and the off-surface drift of 3D Gaussians. To resolve these issues, we propose a method that seamlessly merge 3DGS with the learning of neural SDFs. Our key idea is to more effectively constrain the SDF inference with the multi-view consistency. To this end, we dynamically align 3D Gaussians on the zero-level set of the neural SDF using neural pulling, and then render the aligned 3D Gaussians through the differentiable rasterization. Meanwhile, we update the neural SDF by pulling neighboring space to the pulled 3D Gaussians, which progressively refine the signed distance field near the surface. With both differentiable pulling and splatting, we jointly optimize 3D Gaussians and the neural SDF with both RGB and geometry constraints, which recovers more accurate, smooth, and complete surfaces with more geometry details. Our numerical and visual comparisons show our superiority over the state-of-the-art results on the widely used benchmarks. Project page: https://wen-yuan-zhang.github.io/GS-Pull.

## 1   Introduction

3D scene representations are important to various computer vision applications, such as single or multi-view 3D reconstruction [39, 42, 67, 72], novel view synthesis [1, 71], and neural SLAM [22, 32, 78, 24] etc.. Mesh and point clouds are the most common 3D scene representations, and can be rendered by fast rasterization on GPUs. Instead, more recent neural radiance fields (NeRFs) [49] are continuous scene representations, but it is slow to render NeRFs due to the need of costly stochastic sampling along rays in volume rendering. More recently, 3D Gaussians with different attributes like color and opacity are used as a versatile differentiable volumetric representation [30, 77, 23] for neural rendering through splatting, dubbed 3D Gaussian Splatting (3DGS). It prompts the pros of both NeRFs and point based representations, which achieves both better quality and faster speed in rendering. Although 3D Gaussians can render plausible images, it is still a challenge to reconstruct surfaces based on the 3D Gaussians.

The key challenge comes from the gap between the discrete 3D Gaussians and the continuous geometry representations, such as implicit functions. Besides the discreteness, the sparseness caused by uneven distribution and the off-surface drift make 3D Gaussians even harder to use than

---

[*]The corresponding author is Yu-Shen Liu.

38th Conference on Neural Information Processing Systems (NeurIPS 2024).

scanned point clouds in surface inference. To overcome these obstacles, recent solutions usually add previous volume rendering based reconstruction methods [60, 50] to 3DGS as a complement branch [44, 66, 11], use monocular depth and normal images as priors to bypass the messy and unordered 3D Gaussians [14, 57], or use surface-aligned Gaussians [25, 68, 14] in rasterization to approximate surfaces. However, how to learn continuous implicit representations to recover more accurate, smooth, and complete surfaces with sharp geometry details is still an open question.

To answer this question, we introduce a novel method to infer neural SDFs from multi-view RGB images through 3D Gaussian splatting. We progressively infer a signed distance field by training a neural network along with learning 3D Gaussians to minimize rendering errors through splatting. To more effectively constrain the surface inference with the multi-view consistency, we dynamically align 3D Gaussians with the zero-level set of the neural SDF, and render the aligned 3D Gaussians on the zero-level set by differentiable rasterization. Meanwhile, we update the neural SDF by pulling the neighboring space onto the disk determined by each 3D Gaussian on the zero-level set, which gradually refines the signed distance field near the surface. The capability of seamlessly merging neural SDFs with 3DGS not only get rid of the dependence of costly NeRFs like stochastic sampling on rays but also enables us to access the field attributes like signed distances and gradients during the splatting process, which provides a novel perspective and a versatile platform for surface reconstruction with 3DGS. The key to the 3D Gaussian alignment and neural SDF inference is a differentiable pulling operation which uses the predicted signed distances and gradients from the neural SDF. It provides a way of imposing geometry based constraints on 3D Gaussians besides the RGB based constraints through splatting. Our numerical and visual evaluations on widely used benchmarks show our superiority over the latest methods in terms of reconstruction accuracy and recovered geometry details. Our contributions are listed below,

- We propose to infer neural SDF through splatting 3D Gaussians pulled on the zero-level set, which can more effectively constrain surface inference with the multi-view consistency. This enables to recover more accurate, smooth, and complete surfaces with geometry details.

- We introduce to dynamically align 3D Gaussians to the zero-level set and update the neural SDF through a differentiable pulling operation. To this end, we propose novel loss terms and training strategies to work with the discrete and sparse 3D Gaussians in surface reconstruction.

- We achieve the state-of-the-art numerical and visual results in multi-view based surface reconstruction.

## 2 Related Work

Neural implicit representations have achieved remarkable progress in reconstructing 3D geometry with details [53, 48, 13, 45, 7, 21]. Neural implicit functions can be learned by either 3D supervisions, such as signed distances [53, 13, 41] and binary occupancy labels [48], or 2D supervisions, such as multi-view RGB images [60] and normal images [5]. In the following, we focus on reviewing methods of learning implicit representations from 2D and 3D supervisions separately. Then we provide a detailed discussion on the latest reconstruction methods based on 3D Gaussians.

### 2.1 Learning Implicit Representation from Multi-view Images

Neural radiance fields (NeRFs) [49] have become an essential technology for representing 3D scene through multi-view images. Many of its applications have been explored, resulting in significant advancements in areas such as acceleration [50, 12], dynamic scene [16, 4] and sparse rendering [58, 26]. Besides these applications, extracting accurate surfaces from NeRFs remains a challenge. Mainstream approaches typically design various differentiable formulas to transform the density in radiance fields into implicit representations for volume rendering, such as signed distance function (SDF) [60, 39, 54], unsigned distance function (UDF) [42, 47, 15, 70] and occupancy [52]. With the learned implicit function fields, post-processing algorithms [43, 21, 48] are applied to extract the zero level set to obtain the reconstructed meshes. Following methods introduce different priors from SfM [18, 69] or large-scale datasets [67, 59, 40] to improve the reconstruction performance in large-scale scenes. Recent approaches focus on speeding up the neural rendering procedure, aiming to achieve high-quality meshes and rendering views within a short period of training time. They

propose alternative data structures to replace the heavy MLP framework used in original NeRF, such as sparse voxel grid [17], multi-resolution hash grid [50, 61] and radial basis function [12], or design subtle differentiable rasterization pipelines to achieve real-time rendering [64, 56]. However, these methods still face the trade-off between rendering quality and training speed.

## 2.2 Learning Implicit Representation from Point Clouds

Since DeepSDF [53] and OccNet [48] were proposed, learning implicit representation from point clouds has achieved remarkable results in geometry modeling. These methods use ground truth signed distances and binary occupancy labels calculated from ground truth point clouds as supervisions to learn the implicit representation of shapes. The supervisions can serve as different kinds of global priors [3, 42, 55, 36] and local priors [62, 28, 38, 6], which enables the neural implicit function to better capture geometry details and generalize to unseen shapes during inference. Some other methods infer SDFs without 3D supervisions. They train neural networks to overfit on single point clouds. These methods introduce additional constraints [19, 76], novel ways of using gradients [45, 74, 51, 35, 75], specially designed priors [46, 10, 9] and normals [2, 37, 34] to estimate signed or unsigned distances and occupancy, which use point clouds as a reference.

## 2.3 Surface Reconstruction with 3D Gaussians

3D Gaussian Splatting [30] has become a new paradigm in neural rendering due to its fast rendering speed, intuitive explicit representation and outstanding rendering performance. However, reconstructing accurate surfaces from 3D Gaussian remains a challenge due to the messy, noisy, and unevenly distributed 3D Gaussians. To solve this problem, one kind of approaches involves combining 3D Gaussians with neural implicit surface functions [60, 50] to enhance the performance of both branches, which employs mutual supervisions between the two components [66, 11, 44]. Another kind of approaches encourage the reduction from 3D Gaussians to 2D Gaussians with a series of regularization terms, which ensures the Gaussian primitives to align with the object surfaces [25, 20, 14]. Additionally, some methods introduce additional priors from large-scale datasets [57, 14] or multi-view stereo [63], or use elaborately designed surface extraction algorithms [68, 65] to recover 3D geometry from 3D Gaussians. Although these efforts have achieved improved reconstructions, they are still limited in capturing fine-grained geometry and lack the precise perception of continuous implicit representations. Different from all these mentioned methods, we propose to seamlessly combine 3D Gaussians with the learning of neural SDFs. Our method provides a novel perspective to jointly learn 3D Gaussians and neural SDFs by more effectively using multi-view consistency and imposing geometry constraints.

# 3   Method

**Overview.** We aim to infer a neural SDF $f$ from posed multi-view RGB images $\{v_i\}_{i=1}^{I}$, as shown in Fig. 1. We learn 3D Gaussian functions $\{g_j\}_{j=1}^{J}$ with their attributes like color, opacity, and shape to represent the geometry and color in the 3D scene. Meanwhile, when learning the 3D Gaussians, we introduce novel constraints to infer the continuous surfaces with the neural SDF. We rely on a differentiable pulling operation and the differentiable rasterization to bridge the gap between the discrete Gaussians and the continuous neural SDF, align 3D Gaussians on the zero-level set of the neural SDF, and back propagate the supervision signals from both the rendering errors and other geometry constraints to jointly optimize 3D Gaussians and the neural SDF.

**Neural Signed Distance Function.** We leverage an SDF $f$ to represent the geometry of a scene. An SDF $f$ is an implicit function that can predict a signed distance $s$ at an arbitrary location $q$, i.e., $s = f(q)$. Recent methods usually train a neural network to approximate an SDF from signed distance supervision or infer an SDF from 3D point clouds or multi-view images. A level set is an iso-surface formed by the points with the same signed distance values. The zero-level set is a special level set, which is formed by points with a signed distance of 0. We can use the marching cubes algorithm [43] to extract the zero-level set a a mesh surface. Another character of the zero-level set is that the gradient of the SDF $f$ at query $q$ on the zero-level set, i.e., $\nabla f(q)$, is the normal of $q$.

**3D Gaussian Splatting.** 3D Gaussians have become a vital differentiable volume representation for scene modeling. We can learn a set of 3D Gaussians $\{g_j\}_{j=1}^{J}$, each of which has a set of learnable

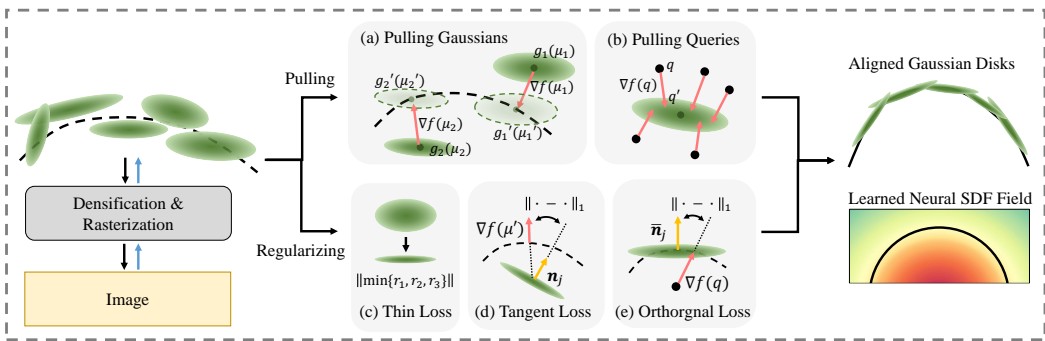

Figure 1: Overview of our method. We **(a)** pull 3D Gaussians onto the zero-level set for splatting, while **(b)** pulling the neighboring space onto the Gaussian disks for SDF inference. To better facilitate this procedure, we introduce three constraints: **(c)** push the Gaussians to become disks; **(d)** encourage the disk to be a tangent plane on the zero-level set; **(e)** constrain the query points to be pulled along the shortest path.

attributes including mean, variances, rotation, opacity, and color. We can render the learnable Gaussians $\{g_j\}$ into RGB images through the volume rendering equation below,

$$\boldsymbol{C}'(u, v) = \sum_{j=1}^{J} \boldsymbol{c}_j * o_j * p_j(u, v) \prod_{k=1}^{j-1} (1 - o_k * p_k(u, v)), \tag{1}$$

where $\boldsymbol{C}'(u, v)$ is the rendered color at the pixel $(u, v)$, $\boldsymbol{c}_i$, $o_i$, and $p_i$ denote the color, the opacity, and the 2D projection of the $j$-th 3D Gaussian, respectively. At a query $q = [x, y, z]$, the probability from the $j$-th 3D Gaussian is $p_j(q) = exp(-0.5 * (q - \mu_j)^T \sum^{-1} (q - \mu_j))$, where $\mu_j$ is the center of the $j$-th Gaussian, and $\sum$ is the covariance matrix.

We can learn these 3D Gaussian functions through a differentiable rasterization. We render 3D Gaussians $\{g_j\}$ into rendered RGB images $v'_i$, and then, optimize the learnable attributes by minimizing the rendering errors to the ground truth observations $v_i$, where $\boldsymbol{C}'(u, v)$ and $\boldsymbol{C}(u, v)$ are the rendered and the GT color values at pixel $(u, v)$, i.e., $\min_{\{g_j\}} ||\boldsymbol{C}'(u, v) - \boldsymbol{C}(u, v)||_2^2$.

**Aligning 3D Gaussians with the Zero-level Set.** Since 3D Gaussian splatting is so flexible in volume rendering, it does not require 3D Gaussians to locate on the geometry surface for good rendering quality. While we expect 3D Gaussians to locate on geometry surface, so that we can more effectively leverage them and multi-view consistency as clues to infer more accurate neural SDFs for reconstruction. To this end, we introduce a differentiable pulling operation to pull 3D Gaussians on the zero-level set of the neural SDF $f$, and then, we render the pulled 3D Gaussians through the splatting.

Specifically, inspired by Neural-Pull [45], we rely on the gradient field of the neural SDF $f$ during the pulling operation. We move each one of the 3D Gaussians $g_j$ using the predicted signed distance $s_j = f(\mu_j)$ and the gradient $\nabla f(\mu_j)$, where $\mu_j$ is the mean value of the 3D Gaussian. As shown in Fig. 1 (a), this pulling operation will turn the 3D Gaussian $g_j$ into a 3D Gaussian $g'_j$ that get projected onto the zero-level set of SDF $f$, where $g'_j$ shares the same attributes with $g_j$ but has a different center $\mu'_j$,

$$\mu'_j = \mu_j - s_j * \frac{\nabla f(\mu_j)}{|\nabla f(\mu_j)|}. \tag{2}$$

**Signed Distance Inference with Pulled 3D Gaussians.** We infer signed distances in the field with pulled 3D Gaussians $\{g'_j\}$. Pulled 3D Gaussians provide a coarse estimation of the surface, which we can use as a reference. One challenge here is that the sparsity and non-uniformly distributed 3D Gaussians do not show a clear geometry clue for surface inference. Although previous methods like NeuralTPS [8] and OnSurfPrior [46] manage to learn continuous implicit functions from sparse points, it is still difficult to recover surfaces from both sparse and non-uniformly distributed points.

To overcome this challenge, we introduce an approach to estimate neural SDFs from sparse 3D Gaussians. Like Neural-Pull [45], we still use a differentiable pulling operation to pull neighboring space onto the surface but we regard the disk established by the shape of a 3D Gaussian as a pulling target, rather than a point, as shown in Fig. 2, which aims for a larger target on surfaces. To this end, we impose constraints not only on the shape of 3D Gaussians but also on the pulling operation. Specifically, we introduce three constraints. The first one constrains 3D Gaussians to be a thin disk. The second constraint encourages the thin disk to be a tangent plane on the zero-level set. The third constraint pushes queries to get pulled onto the thin disk along the normal of the Gaussian.

The first constraint adds penalties if the smallest variance among the three variances of a 3D Gaussian $g_j$ is too large, as shown in Fig. 1 (c). Thus, the loss for a thin disk Gaussian is listed below,

$$L_{Thin} = \|\min\{r_1, r_2, r_3\}\|_1, \tag{3}$$

where $r_1$, $r_2$, and $r_3$ are variances along the three axes. Flattening a 3D Gaussian ellipsoid into a disk was first introduced in NeuSG [11] and has become a consensus in recent Gaussian reconstruction works [25, 14]. The motivation is that 2D planar disk primitives are more suitable for surface representation, making it easier to apply alignment constraints. Additionally, we can naturally use the direction pointing along the axis with the minimum variance $\bar{r} = \min\{r_1, r_2, r_3\}$ to represent the normal $n_j$ of the Gaussian $g_j$.

Based on the thin disk shape of Gaussians, the second constraint encourages the pulled Gaussians $\{g'_j\}$ to be the tangent plane on the zero-level set, as shown in Fig. 1 (d). What we do is to align the normal $n_j$ of a Gaussian $g_j$ with the normal at the center $\mu'_j$ of the pulled Gaussian $g'_j$ on the zero-level set. We use the gradient $\nabla f(\mu'_j)$ of the neural SDF at $\mu'_j$ as the expected normal here. Hence, we align the normal $n_j$ of a Gaussian with the normal $\nabla f(\mu'_j)$ on the zero-level set,

$$L_{Tangent} = 1 - \left| \frac{\nabla f(\mu'_j)}{|\nabla f(\mu'_j)|} \cdot n_j \right| \tag{4}$$

With the disk-like Gaussians located on the tangent plane, we introduce to sense the signed distance field by pulling randomly sampled queries on the Gaussian disks, as shown in Fig. 1 (b). Turning the pulling target from a point [45, 8] into a plane is based on the observation that the 3D Gaussian function with a boundary can cover the surface more completely although their centers $\{\mu_1, ..., \mu_j\}$ which are sparse and non-uniformly distributed. Thus, we expect the operation can pull a query onto a Gaussian disk plane. Fig. 2 demonstrates the improvement of pulling queries onto their nearest Gaussian disk planes over the nearest Gaussian centers. The comparisons show that pulling onto the disk plane can improve the robustness to the sparsity and non-uniformly Gaussian distribution. With learned Gaussian centers, pulling queries to centers can

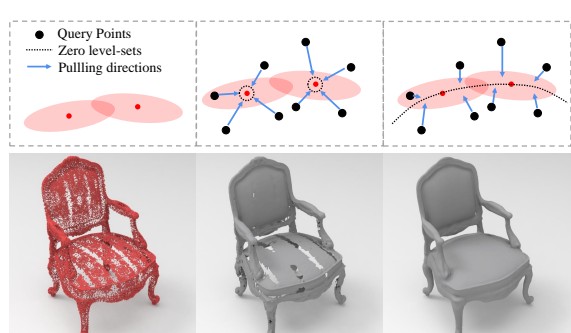

Gaussian Centers    Pulled to Centers    Pulled to Disks

Figure 2: Comparison of pulling Gaussians to centers and to disks. The former tends to overfit sparse Gaussian centers, resulting in incomplete meshes. We address this issue by pulling queries onto disk planes.

not recover the smooth and continuous geometry in areas where almost no Gaussian centers appear. While pulling queries to the Gaussian disk plane can recover more accurate and complete surfaces since the disk established by the learned variance of Gaussian functions can mostly cover the gap.

Specifically, at a query $q$, we pull it onto the zero-level set using a similar way in Eq. 2, i.e., $q' = q - s * \nabla f(q)/|\nabla f(q)|$. To encourage the query to get pulled onto the nearest pulled Gaussian disk, we maximize its probability of belonging to its nearest pulled Gaussian $\bar{g}$ which is determined in terms of the distance between $q$ and the Gaussian center $\bar{\mu}$,

$$L_{Pull}(q'; \bar{\mu}) = e^{-1/2*(q'-\bar{\mu})^T \Sigma^{-1}(q'-\bar{\mu})}, \quad \bar{g} = \underset{\{g'_j\}}{\arg\min} \, ||\mu'_j - q||_2^2. \tag{5}$$

We minimize the negative logarithm of the probability in our implementation. Moreover, we expect the pulling can follow a direction orthogonal to the disk plane, which leads to the minimum moving distance conform to the definition of signed distances. To this end, we impose another constraint on the gradient to ensure that the pulling can follow a path with the minimum distance to the nearest pulled Gaussian disk, as shown in Fig. 1 (e),

$$L_{Othorgnal} = 1 - \left| \frac{\nabla f(q)}{|\nabla f(q)|} \cdot \bar{n}_j \right|, \tag{6}$$

where the constraint aligns the gradient at query $q$ and the normal $\bar{n}_j$ of the pulled Gaussian disk $\bar{g}$.

**Rendering.** We also render the pulled Gaussians $\{g'_j\}$ into images through splatting to add penalties on rendering errors, where $\{g'_j\}$ are Gaussians pulled onto the zero-level set from the Gaussians $\{g_j\}$ by the neural SDF $f$ in Eq. 2. Each pair of $g_j$ and $g'_j$ shares the same attributes expect the center location. The rendering error combines an $L_1$ term and a D-SSIM term between rendered images $\{v'_i\}_{i=1}^I$ and ground truth ones $\{v_i\}_{i=1}^I$, following original 3DGS [30],

$$L_{Splatting} = 0.8 \cdot L_1(v'_i, v_i) + 0.2 \cdot L_{D-SSIM}(v'_i, v_i). \tag{7}$$

**Loss Function.** We optimize attributes of Gaussians $\{g_j\}$ and the parameters of neural SDF $f$ by the following objective function, where $\alpha$, $\beta$, $\gamma$, and $\delta$ are balance weights.

$$\underset{\{g_j\}, f}{\min} \, L_{Splatting} + \alpha L_{Thin} + \beta L_{Tangent} + \gamma L_{Pull} + \delta L_{Othorgnal}. \tag{8}$$

**Implementation Details.** Our code is build upon the source code released by 3DGS [30]. Similar to [68], we make some changes to 3DGS's densification strategy. The first one is to initialize the newly cloned Gaussians around the original Gaussians rather than at the same positions. The second one is to encourage 3DGS to split larger Gaussians into smaller ones more frequently. These strategies aim to increase the number of primitives and to avoid underfitting in textureless areas. Regularization parameters are set to $\alpha$=100, $\beta$=0.1, $\gamma$=1, $\delta$=0.1. We optimize our model for a total of 15k iterations. We stop densification and incorporate the pulling and constraints at 7k iterations. The SDF network is implemented as an MLP with 8 layers, 256 hidden units and ReLU activation function, and initialized as a sphere, following [45]. The parameters of the SDF network shares the same optimizer as that of 3D Gaussians. All the experiments are conducted on a single NVIDIA 3090 GPU.

## 4 Experiments

### 4.1 Experiment Settings

**Evaluation Metrics and Datasets.** We evaluate the performance of our method on widely adopted datasets including both object-level and large-scale ones, including DTU [27], Tanks and Temples (TNT) [31] and Mip-NeRF 360 (M360) [1]. To evaluate the accuracy of the reconstructed meshes, we use Chamfer Distance (CD) on DTU and F-score on TNT, using the official evaluation script. To evaluate the rendering quality in real-scene datasets, we report PSNR, SSIM and LPIPS in evaluations on M360.

**Baselines.** We compare our geometry reconstruction accuracy with the state-of-the-art 3DGS based reconstruction methods, including SuGaR [20], DN-Splatter [57], GaussianSurfels [14] and 2DGS [25]. For real-world scenes which do not have ground truth meshes for evaluations, we compare the rendering quality with state-of-the-art neural rendering methods, including Instant-NGP [50], Mip-NeRF 360 [1] and BakedSDF [64].

**Surface Extraction.** An advantage of our approach over the latest methods is the simplicity of extracting surfaces. Different from methods like SuGaR [20] and GauS [65] which introduce

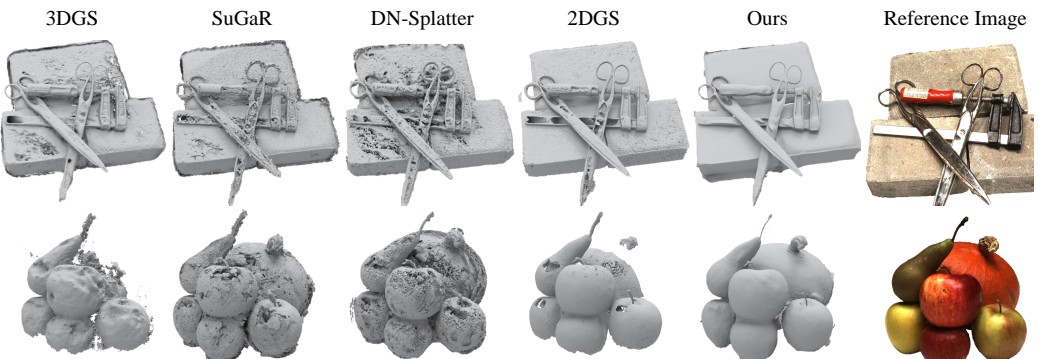

Figure 3: Visual comparisons on DTU dataset.

Table 1: Numerical comparisons in terms of CD on DTU dataset. Best results are highlighted as 1st, 2nd and 3rd.

| Methods | 24 | 37 | 40 | 55 | 63 | 65 | 69 | 83 | 97 | 105 | 106 | 110 | 114 | 118 | 122 | Mean | Time |
|---|---|---|---|---|---|---|---|---|---|---|---|---|---|---|---|---|---|
| NeuS [60] | 1.00 | 1.37 | 0.93 | 0.43 | 1.10 | 0.65 | 0.57 | 1.48 | 1.09 | 0.83 | 0.52 | 1.20 | 0.35 | 0.49 | 0.54 | 0.84 | ~9h |
| 3DGS [30] | 2.14 | 1.53 | 2.08 | 1.68 | 3.49 | 2.21 | 1.43 | 2.07 | 2.22 | 1.75 | 1.79 | 2.55 | 1.53 | 1.52 | 1.50 | 1.96 | 15.1m |
| SuGaR [20] | 1.47 | 1.33 | 1.13 | 0.61 | 2.25 | 1.71 | 1.15 | 1.63 | 1.62 | 1.07 | 0.79 | 2.45 | 0.98 | 0.88 | 0.79 | 1.33 | 1.6h |
| DN-Splatter [57] | 1.60 | 2.03 | 1.42 | 1.44 | 2.37 | 2.11 | 1.62 | 1.95 | 1.88 | 1.48 | 1.63 | 1.82 | 1.20 | 1.50 | 1.40 | 1.70 | 31.2m |
| GSurfels [14] | 0.66 | 0.93 | 0.54 | 0.41 | 1.06 | 1.14 | 0.85 | 1.29 | 1.53 | 0.79 | 0.82 | 1.58 | 0.45 | 0.66 | 0.53 | 0.88 | 10.9m |
| 2DGS [25] | 0.48 | 0.91 | 0.39 | 0.39 | 1.01 | 0.83 | 0.81 | 1.36 | 1.27 | 0.76 | 0.70 | 1.40 | 0.40 | 0.76 | 0.52 | 0.80 | 20.5m |
| Ours | 0.51 | 0.56 | 0.46 | 0.39 | 0.82 | 0.67 | 0.85 | 1.37 | 1.25 | 0.73 | 0.54 | 1.39 | 0.35 | 0.88 | 0.42 | 0.75 | 21.8m |

Table 2: Numerical comparisons on Tanks And Temples dataset. Best results are highlighted as 1st, 2nd and 3rd.

| Methods | Barn | Caterpillar | Courthouse | Ignatius | Meetingroom | Truck | Mean | Time |
|---|---|---|---|---|---|---|---|---|
| NeuS [60] | 0.29 | 0.29 | 0.17 | 0.83 | 0.24 | 0.45 | 0.38 | ~12h |
| 3DGS [30] | 0.13 | 0.08 | 0.09 | 0.04 | 0.01 | 0.19 | 0.09 | 20.5m |
| SuGaR [20] | 0.14 | 0.16 | 0.08 | 0.33 | 0.15 | 0.26 | 0.19 | 2.1h |
| DN-Splatter [57] | 0.15 | 0.11 | 0.07 | 0.18 | 0.01 | 0.20 | 0.12 | 54.9m |
| GSurfels [14] | 0.24 | 0.22 | 0.07 | 0.39 | 0.12 | 0.24 | 0.21 | 15.1m |
| 2DGS [25] | 0.41 | 0.23 | 0.16 | 0.51 | 0.17 | 0.45 | 0.32 | 39.4m |
| Ours | 0.60 | 0.37 | 0.16 | 0.71 | 0.22 | 0.52 | 0.43 | 37.6m |

specially designed algorithms and take a long time for extracting surfaces, we adopt the marching cubes algorithm [43] to extract mesh surfaces with the learned neural SDF $f$. For small scale scenes, we use a resolution of 800 to extract surfaces, while we split large scale scenes into parts, each of which gets reconstructed with a resolution of 800 to bypass the limitation of our computational resources.

### 4.2 Comparisons

**Comparisons on DTU.** We report accuracy of reconstructed meshes and training time against baselines on DTU dataset in Tab. 1. Our method outperforms all Gaussian-based reconstruction methods in terms of Chamfer Distance. Our method achieves comparable training time to the state-of-the-art Gaussian-reconstruction method 2DGS [25] but gains better reconstruction accuracy than 2DGS. The visualization results in Fig. 3 highlight the advantages of our method. By employing alignment constraints and pulling operations between the 3D Gaussians and the neural SDF field, we can reconstruct significantly smoother and more complete surfaces than the baselines.

**Comparisons on TNT.** We further evaluate our method using more challenging large-scale unbounded scenes on TNT dataset. Numerical comparisons in Tab. 2 show that we achieve higher F-score compared to baseline methods, even surpassing NeuS, which however takes about 12 hours to fit a scene. Notably, as the scene scale increases, the number of Gaussian primitives increases rapidly, causing the adjusted CUDA rasterization kernel of 2DGS to consume more time for rendering. In contrast, since our rasterization kernel is based on 3DGS, it is less sensitive to the number of Gaussians, which enables us to learn 3D Gaussians faster than 2DGS. We provide visual comparisons

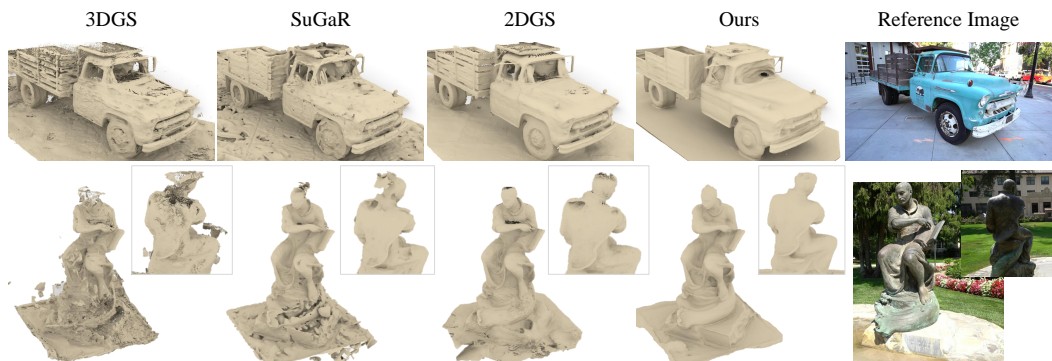

Figure 4: Visual comparisons on Tanks and Temples dataset.

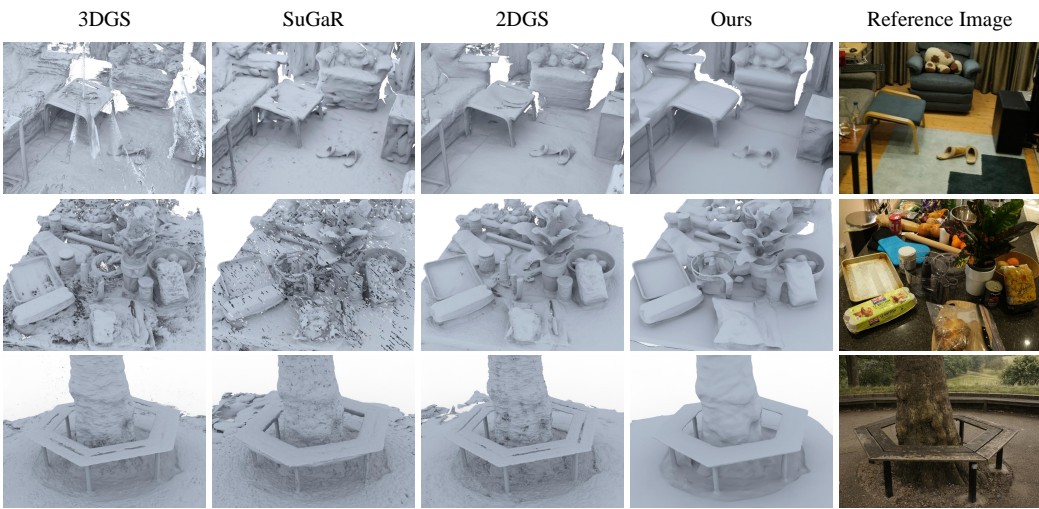

Figure 5: Visual comparisons on Mip-NeRF 360 dataset.

in Fig. 4. Here we crop the reconstructed meshes to show the foreground objects that are of primary interest, as captured by the cameras. Please refer to the appendix for the reconstruction results of the background regions. The visual comparisons demonstrate that we can reconstruct more complete and smooth object surfaces, such as the ground, the truck's hood and the statue's left shoulder.

**Comparisons on MipN-eRF 360.** We further evaluate our method in neural rendering for novel view synthesis on MipNeRF 360 dataset. We report the numerical comparisons in Tab. 3. Our competitive results against the state-of-the-art novel view synthesis methods indicate that our method is able to impose effective geometric constraints without compromising rendering quality. This provides a promising solution for learning continuous distance fields from discrete 3D Gaussians. Visual comparisons of mesh reconstructions are shown in Fig. 5, which demonstrate that our method is able to recover more smooth and complete surface by more effectively using the multi-view consistency.

Table 3: Quantitative evaluations of rendering quality on Mip-NeRF 360 [1] dataset. Best results are highlighted as 1st , 2nd and 3rd .

|  | Indoor Scene | | | Outdoor Scene | | |
|---|---|---|---|---|---|---|
|  | PSNR↑ | SSIM↑ | LPIPS↓ | PSNR↑ | SSIM↑ | LPIPS↓ |
| NeRF[49] | 26.84 | 0.790 | 0.370 | 21.46 | 0.458 | 0.515 |
| Instant-NGP [50] | 29.15 | 0.880 | 0.216 | 22.90 | 0.566 | 0.371 |
| MipNeRF 360 [1] | 31.72 | 0.917 | 0.180 | 24.47 | 0.691 | 0.283 |
| BakedSDF [64] | 27.06 | 0.839 | 0.258 | 22.47 | 0.585 | 0.349 |
| 3DGS [30] | 30.99 | 0.926 | 0.199 | 24.24 | 0.705 | 0.283 |
| SuGaR [20] | 29.44 | 0.911 | 0.216 | 22.76 | 0.631 | 0.349 |
| 2DGS [25] | 30.39 | 0.924 | 0.183 | 24.33 | 0.709 | 0.284 |
| Ours | 30.78 | 0.925 | 0.182 | 23.76 | 0.703 | 0.278 |

Table 4: Ablation studies on DTU dataset.

| | Pulling | | Constraint Terms | | | Mesh Extractions | | |
|---|---|---|---|---|---|---|---|---|
| Methods | Pulled to centers | w/o Pull GS | w/o $L_{Thin}$ | w/o $L_{Tan}$ | w/o $L_{Oth}$ | TSDF | Poisson | Full model |
| CD↓ | 0.85 | 0.90 | 0.78 | 0.82 | 0.79 | 1.41 | 0.79 | 0.75 |

## 4.3 Ablation Studies

In this section, we conduct ablation studies on the key techniques of our method to demonstrate their effectiveness. The full quantitative results are reported in Tab. 4, which are conducted on all scenes in DTU dataset [27].

**Pulling Operations.** We first examine the effect of pulling Gaussians onto the zero-level set, as reported in Tab. 4 ("w/o Pull Gaussians" vs. "Ours"). The original 3DGS tends to produce floating ellipsoids near the object surfaces to overfit the training views. By pulling the Gaussians to the zero-level set of the SDF field, the Gaussians are consistently distributed on the surface. As shown in Fig. 6a, after getting pulled onto the zero-level set, the Gaussian centers are distributed on a thin layer of the object surface, thus achieving an accurate geometry estimation. Meanwhile, we pull neighboring space onto Gaussian disks to learn neural SDFs. Comparing to NeuralPull [45] which pulls query points to centers, we innovatively pull query points to Gaussian disks, which bridge the gap between continuous SDF field and sparse Gaussian distributions, as highlighted in Fig. 2 and Tab. 4 ("Pulled to centers" vs. "Ours").

**Constraint Terms.** We further explore the effect of our constraint terms, as reported in Tab. 4 ("Constraint Terms"). Our full model provides the best performance when applying all constraint terms. The orthogonal loss helps to learn a more regularized SDF field, while the thin loss and tangent loss provide constraints to align the orientation of Gaussian disks with the gradient of neural SDF on the zero-level sets, resulting in a good normal field and a reconstructed mesh, as shown in Fig. 6b, 6c.

**Mesh Extraction.** We also report the reconstruction accuracy using TSDF fusion

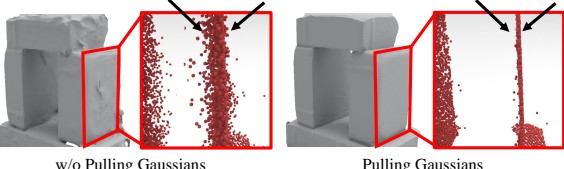

(a) Visualization of Gaussian centers with or without pulled onto zero-level set. We are able to obtain consistent and smooth Gaussian distributions by pulling operation.

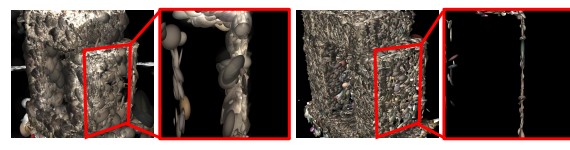

(b) Comparisons between Gaussian ellipsoids learned by original 3DGS and Gaussian disks learned by our method.

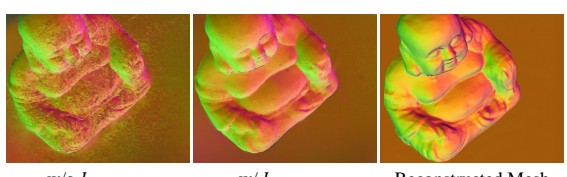

(c) Qualitative ablation studies for Tangent loss.

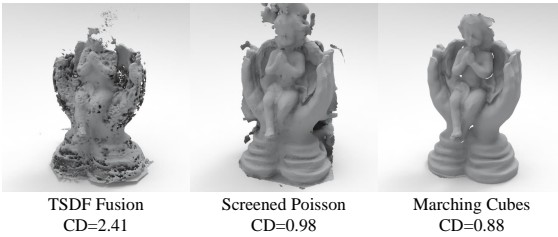

(d) Comparisons of different mesh extraction methods.

and screened Poisson reconstruction [29], as shown in Fig. 6d and Tab. 4 ("Mesh Extractions"). For TSDF fusion, we render depth maps and fuse them using a voxel size of 0.004 and truncation threshold as 0.02, the same as 2DGS [25]. For screened Poisson, we use the Gaussian centers and normals as input. Unlike 2DGS [25] and GSurfels [14] which incorporate rendered depth into the differentiable rasterization pipeline, we do not directly optimize depths, resulting in noisy depth maps and unsatisfactory reconstruction results. However, since the positions and normals of the Gaussians are well optimized through our approach, screened Poisson reconstruction can achieve relatively good results.

**Gradient Constraint.** We follow Neural-Pull [45] to use normalized SDF gradient for pulling operation. We report the result with an additional Eikonal term [19] to explicitly constrain the gradient length, as shown in Fig. 7. The result is significantly degenerated because that Neural-Pull depends on both predicted SDF values and gradient directions to optimize the SDF field. It makes the optimization even more complex when adding additional constraint on the gradient length.

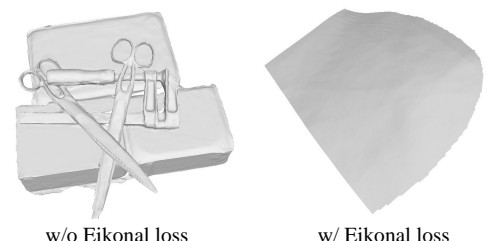

w/o Eikonal loss       w/ Eikonal loss

Figure 7: Visualization of the effect of eikonal loss.

## 5 Conclusion

We propose a method to learn neural SDFs for multi-view surface reconstruction with 3D Gaussian splatting. Our results show that we can more effectively leverage multi-view consistency to recover more accurate, smooth, and complete surfaces with geometry details by rendering 3D Gaussians pulled on the zero-level set. To this end, we dynamically align 3D Gaussians to the zero-level set and update neural SDFs through both differentiable pulling and splatting for both RGB and geometry constraints. Our methods successfully refine the signed distance field near the surface in a progressive manner, leading to plausible surface reconstruction. Our ablation studies justify the effectiveness of our novel modules, loss terms, and training strategies. Our evaluations show our superiority over the latest methods in terms of accuracy, completeness, and smoothness.

## 6 Acknowledgement

This work was supported by National Key R&D Program of China (2022YFC3800600), and the National Natural Science Foundation of China (62272263, 62072268), and in part by Tsinghua-Kuaishou Institute of Future Media Data.

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

# A Appendix

## A.1 Background Reconstruction

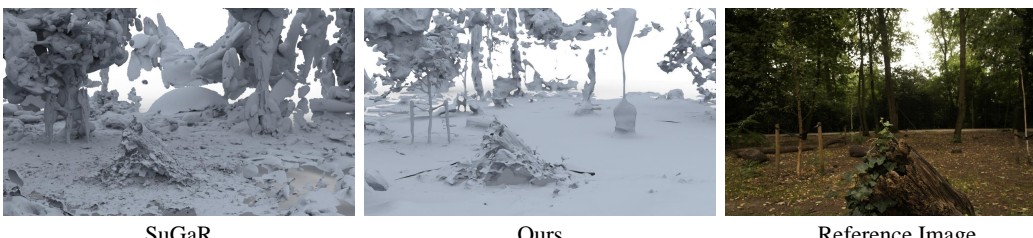

SuGaR            Ours            Reference Image

Figure 8: Visualization of reconstructed backgrounds.

Since our SDF field was learned by fitting Gaussian ellipsoids, it can infer implicit surfaces at any location where Gaussians are distributed. Therefore, our method has the same capability to reconstruct backgrounds as methods like TSDF fusion, as shown in Fig. 8. Current works generally utilize screened Poisson or TSDF fusion to reconstruct meshes [25, 20] and tend to reconstruct large sky spheres in the background. Our method learns neural SDFs and utilize marching cubes to reconstruct mesh, which avoid such bad cases.

## A.2 Theoretical Analysis

We provide a theoretical analysis here to demonstrate the advantage of pulling queries onto disks compared to pulling queries onto centers. We provide a visual comparison of the two strategies in Fig. 9, showcasing the changes of the loss function and the loss gradients as the query point approaches the Gaussian center. As the query point gets closer to the Gaussian center, the loss function of "pulling to centers" decays at a constant rate, and the gradient of the loss remains constant. In contrast, for "pulling to disks", the loss function decreases quadratically, and the gradient of the loss gradually diminishes. This means that under the influence of the disk loss, as the query point approximates the center, the received gradient becomes smaller, reducing the driving force that pushes the query point towards the center. In other words, the disk loss has a higher "tolerance" for the query point not being pulled to the center. This explains why we can learn a continuous field from a sparse and non-uniformity distribution of

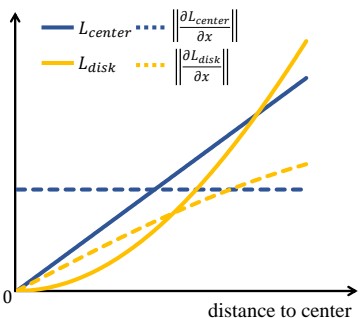

Figure 9: Visualization of loss and gradient between *Pulling to centers* and *Pulling to disks* with the distance of a query point to the Gaussian center.

Gaussian ellipsoids using the disk loss, whereas the center loss would lead to the SDF field overfitting to every Gaussian center.

## A.3 Limitations & Future Works

While our method successfully recovers accurate appearance and geometry reconstruction for a wide range of objects and scenes, it also has several limitations. Firstly, the neural SDF is seamlessly integrated with Gaussian ellipsoids, making it difficult to avoid the inherent drawbacks of original 3D Gaussians, such as the lack of transparent objects and areas with strong specular reflections. Secondly, although we address the issue of learning a continuous SDF field from sparse and non-uniformly distributed Gaussian ellipsoids by pulling query points to disks,

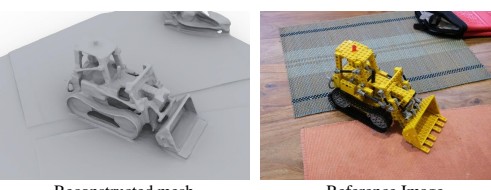

Reconstructed mesh         Reference Image

Figure 10: Failure case. This is because the SDF network cannot accurately capture high-frequency details due to the smooth characteristic of MLPs.

our method shows limited performance in ex-
tremely sparse areas. In very distant regions of unbounded scenes or areas with colors similar to the background color, where 3DGS reconstructs no ellipsoids or only a few ellipsoids, our method tends to produce holes. Thirdly, due to the continuous and smooth characteristics of MLPs, our SDF tends to capture the low-frequency features of objects, making it difficult to reconstruct high-frequency details. A failure case is shown in Figure. 10, where we can reconstruct the very smooth tablecloth but fail to recover the details of the lego. There are two potential solutions for this issue in the future: one is to enhance the representation capability of the SDF by integrated with latest implicit representation learning methods, such as BACON [41] and GridPull [7]; the other one is to dig into the capabilities of TSDF fusion and screened Poisson in reconstructing our SDF field, which have the ability to reconstruct arbitrary resolution details.

## A.4 More Results

We provide additional reconstruction results in Fig. 11, 12, 13, which further justifies the superiority of our method. We notice that there are some holes on the flowerbed area in Fig. 13. This is due to the overly complex geometric structures and a lack of view covering, thus emitting a significant under-fitting issue. This results in a set of extremely sparse, huge, and unevenly distributed Gaussians, which makes Gaussians are thick ellipsoid like shape rather than relatively thin plans, leading to poor sense of surface. Although these huge Gaussians may work well in rendering, but they cannot recover any geometry covered by them. How to control the size of Gaussians for SDF inference could be an interesting future work direction.

We also visualize the error maps on meshes obtained by 2DGS and ours in Fig. 14, which highlights our superiority in terms of the accuracy of extracted surfaces. The surfaces learned by 2DGS are usually fat and a little bit drift away from ground truth surfaces, although their meshes seem to show more details. Our method is able to capture more accurate surfaces by using 3D Gaussians pulled onto the zero-level set and pulling query points onto Gaussian disks at the same time, leading to much more accurate zero-level set.

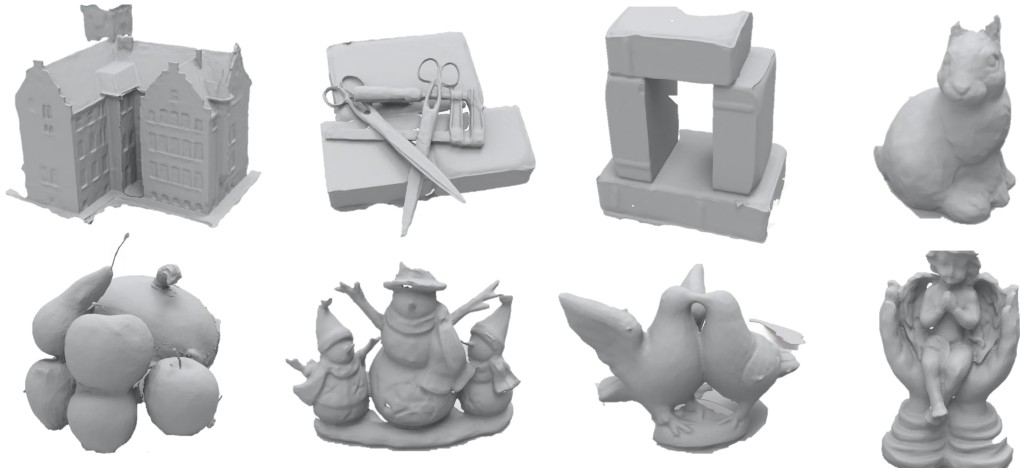

Figure 11: More visualization results on DTU dataset.

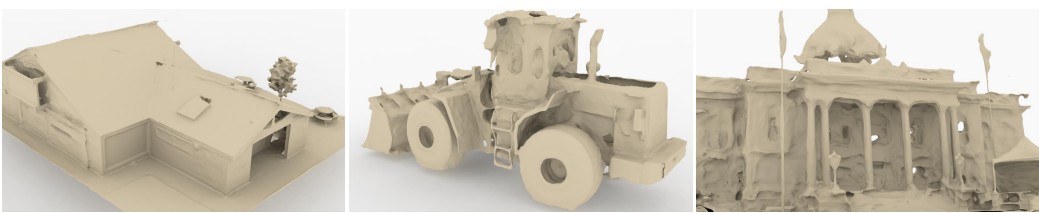

Figure 12: More visualization results on TNT dataset.

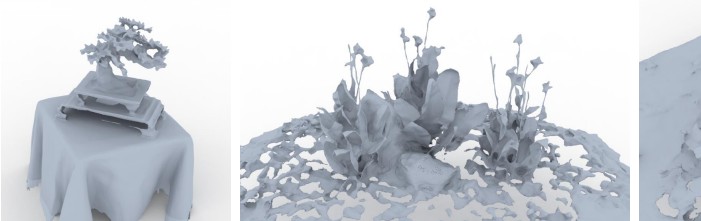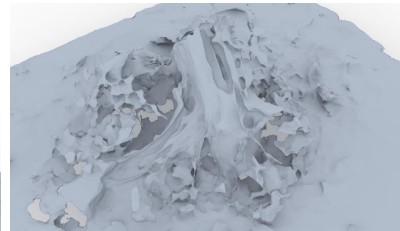

Figure 13: More visualization results on MipNeRF 360 dataset.

2DGS           Ours

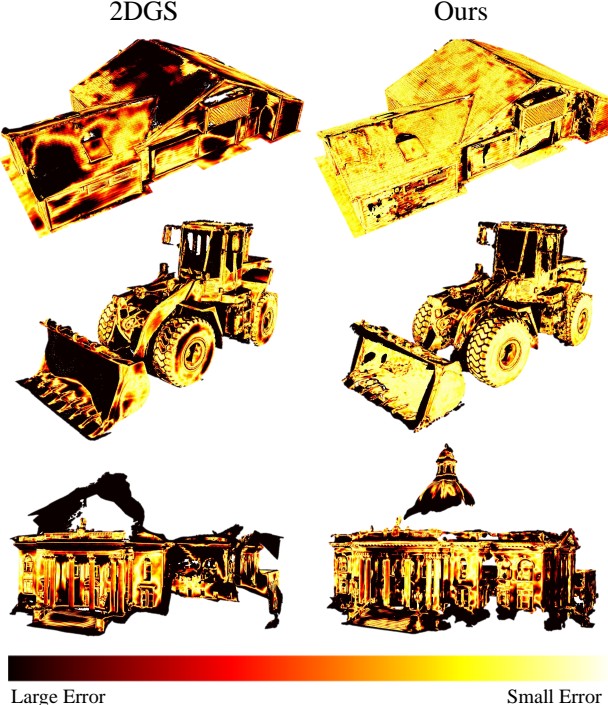

Large Error              Small Error

Figure 14: Error maps between 2DGS and our method on TNT dataset.

### A.5 Discussion

**About open surfaces.** Since there are lots of non-closed surfaces in large-scale scenes, a natural solution is to learn an unsigned distance field to reconstruct open structures [70, 73, 33]. However, extracting the zero-level set from UDF as a mesh surface is still a challenge, resulting in artifacts and outliers on the reconstructed meshes. We report the result of learning UDFs instead SDFs in Fig. 15, which shows the shortcomings of UDF learning. To avoid the influence of double-layer

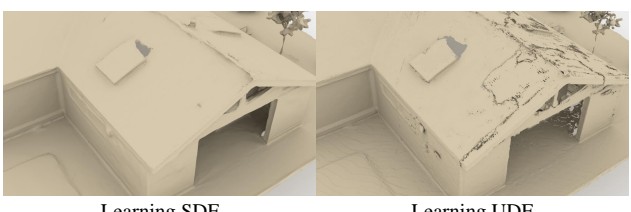

Learning SDF       Learning UDF

Figure 15: Visualization of surfaces reconstructed by SDF and UDF.

surfaces on evaluation accuracy under the SDF settings, we practically delete the back faces according to the visibility of each face under each camera view. Through this way, we can accurately reconstruct open structures with single-layer surfaces.

