# OpenReview forum: "Neural Signed Distance Function Inference through Splatting 3D Gaussians Pulled on Zero-Level Set"
_NeurIPS.cc/2024/Conference — NeurIPS 2024 poster_

### Official Review · Reviewer_ZhFP · 2024-07-10

**Soundness:** 2
**Presentation:** 2
**Contribution:** 3
**Rating:** 5
**Confidence:** 4

**Summary:**

This paper aims to infer neural signed distance functions (SDF) for Gaussian Splatting. To this end, this paper introduce an MLP to represent  the SDF. To learn SDF from sparse and non-uniform Gaussian points, this paper introduces a differentiable pulling operation from Neural-Pull to align Gaussians on the zero-level set of the SDF and updates the SDF by pulling neighboring space to the pulled 3D Gaussians. This paper designs tangent loss, pull loss and orthogonal loss to encourage the above operations. Experiments on DTU and Tanks and Temples datasets show that the proposed method achieves SOTA reconstruction performance.

**Strengths:**

* This work combines a differentiable pulling operation with Gaussian Splatting for surface reconstruction, which helps to learn neural SDF for GS. This is interesting.
* The approach designs the tangent loss to encourage the pulled Gaussians to be the tangent plane on the zero-level set.
* To tackle the sparsity and non-uniform Gaussian distribution, the method proposes to pull randomly sampled points to disks.
* The method achieves promising surface reconstructions on Tanks and Temples dataset.

**Weaknesses:**

* The proposed method cannot reconstruct geometric details well as shown in Figures 4, 13 and 14.
* The ablation results in Table 4 are not consistent with the results in Table 2.
* The ablation study is bad and confusing. The quantitative results are conducted on Tanks and Temples but the qualitative results are conducted on DTU.
* The rendering evaluation setting is not consistent with prior works, such as 2DGS and GOF. Using lower resolution training images can improve rendering results. However, the training setting for NeRF-based methods still use the original setting.

**Questions:**

* Gaussian splatting will generate many points that are far away from surfaces, how does the pulling operation tackle these points?
* When encouraging the thin disk to be a tangent plane on the zero-level set, why not make $f(\mu'_j)\approx0$?
* For Table 4, can you explain the difference from Table 2 in terms of the performance of full model? Moreover, can you explain the detailed operations of w/o Pull GS? I wonder how to train the SDF without Pull GS.
* As the proposed method uses the densification from GOF, it is better compare the rendering performance of proposed method with GOF.
* How many Gaussian points and random points are sampled to train SDF during each optimization?

**Limitations:**

The authors have discussed the limitations and their future works.

---

> ### Author Rebuttal · Authors · 2024-08-07
>
> **1. Reconstructing details.** Since we use the marching cubes algorithm to extract the zero-level set as a mesh surface, our results is limited by the resolution used by the marching cubes. While 2DGS uses multi-view depth images to reconstruct meshes with Poisson surface reconstruction, which achieves higher resolution and recover more geometry details. We additionally use the same way of reconstruction as 2DGS, and show visual comparison in **Fig. F** in our rebuttal PDF, where we can see that our reconstruction show more geometry details than 2DGS.
>
> **2. Results between Tab.2 and Tab.4.** Thanks for pointing this out. Before the deadline, we updated our results frequently, but we forgot to update the latest results everywhere in the paper. We wrongly reported an older version of ablation study, which was conducted on the first four scenes in TNT dataset. We will update the ablation study with the final version, which was conducted on all of the TNT scenes, as reported in **Tab. B** in the rebuttal PDF.
>
> **3. About ablation study.** The TNT scenes are large and complex, therefore we believe the quantitative results on TNT dataset are more convincing than that of DTU dataset. However, the qualitative differences are more distinct on small object scenes such as DTU dataset, therefore we chose DTU dataset for visualization. Due to the time limit, we select the first four scenes in DTU dataset to conduct additional ablation study to demonstrate the effectiveness of our method, as shown in **Tab. C** in our rebuttal PDF. We will update the results with all scenes of DTU in our revision.
>
> **4. Rendering evaluation setting.** For fair comparison, we directly borrowed the results of all baseline methods from Table 4 in 2DGS paper. Due to the limit of memory, current methods, including NeRF-based and 3DGS-based, coherently train outdoor scenes with a down-sampling factor of 4 and train indoor scenes with a factor of 2. Our experiments keep the same settings with baseline methods. We will clarify this in our revision.
>
> **5. About far away points.** Indeed, most of the Gaussian points are distributed around the ground truth surfaces, which can be seen as a noisy point cloud. The Gaussians that are significantly far away from the surfaces usually have small opacity values. In our implementation, we practically utilize a filtering strategy before pulling operation to filter out the Gaussians that have too small opacities, similar as what SuGaR does at 9000 epoch. Through this strategy, almost all of the Gaussian points are distributed around the surfaces. If there are several points that are still far away, the loss gradient of these points will be averaged by other points and they will be ignored by the learned SDF.
>
> **6. why not make $f(\mu_j’)=0$.** Our preliminary experiments show that directly constraining the Gaussians to be the zero-level set will hurt the learning of SDF, as shown in **Fig. G** in our rebuttal PDF. The reason is that the signed distance field has large uncertainty during the optimization, and the pulled queries produced by inaccurate pulling will lead to inaccurate zero-level set. Directly using a hard constraint will make the optimization inefficient. Our results show that implicitly aligning Gaussian disks to the zero-level set resolves this problem.
>
> **7. About w/o Pull GS.** We train SDF by pulling randomly sampled queries to their nearest Gaussians on the zero-level set. The setting of “w/o Pull GS” means that we do not pull Gaussians onto the zero-level set and just pull randomly sampled queries onto their nearest Gaussians. The full model with pulling GS means that we align the Gaussians with zero-level set of SDF, which means that the positions of Gaussians are encouraged to move towards the zero-level set.
>
> **8. Comparison with GOF.** Our method shows comparable rendering results with GOF, as reported in **Tab. D** in our rebuttal PDF.
>
> **9. Number of Gaussian points.** We sample 100,000 points to train SDF in each epoch. Because we need to calculate the nearest Gaussian for each sampling point, the number of sampled Gaussian points per epoch is also 100,000.

---

> ### Comment · Reviewer_ZhFP · 2024-08-11
>
> Thank you for the response. I have also read the other comments and responses. I have some following questions.
> **1. Reconstruction details.** 2DGS uses TSDF fusion to reconstruct meshes. However, the method combines Poisson surface reconstruction to reconstruct meshes from Gaussian points to show the details. It is unfair. Moreover, it is easy to extract higher resolution meshes from SDF, as shown in MonoSDF [1]. Can you show higher-resolution meshes on DTU or TNT dataset? In fact, DTU is a small-scale object-centric dataset. However, the proposed method cannot reconstruct details well on DTU dataset.
> Yu, Zehao, et al. "Monosdf: Exploring monocular geometric cues for neural implicit surface reconstruction." Advances in neural information processing systems 35 (2022): 25018-25032.
> **3. About ablation study.** Since TNT dataset is more challenging, it is better to show ablation qualitative results on this dataset.  Since the the proposed method cannot handle details well, I would like to know how the proposed strategies improve surface reconstruction on the TNT dataset.
> **6. Why not make $f(\mu'_j)\approx0$?** In fact, I wonder if the **$f(\mu'_j)\approx0$** operation can help reconstruction instead of $f(\mu'_j)=0$. Moreover, can you try this using the filtered Gaussian points as you mentioned in *5. About far away points.*
> **8. Comparison with GOF.** The results shown in Table D are not consistent with those in the main paper. Can you check them carefully?
> In addition, since the proposed method is efficient, why only using 4 scenes to conduct ablation studies on DTU dataset? I am curious whether the results in Table C are convincing.

---

> > ### Author Response · Authors · 2024-08-11
> >
> > Dear reviewer ZhFP,
> > Thanks for your comments. We are happy to answer your questions. Unfortunately, we are not allowed to either 1) show you any reconstruction visualization or visual comparison during this reviewer-author discussion period or 2) provide you any URL towards these visual results, according to the rebuttal policy.
> >
> > 1. Reconstructing details.
> >
> > We are sorry for missing the details of 2DGS, since we confused 2DGS and GaussianSurfels, one uses TSDF to extract the zero-level set while the other uses Poisson surface reconstruction. We just ran TSDF fusion on our results to compare, and got comparable results to our results obtained by Poisson surface reconstruction. This is because we use the same depth as input, and the resolution used either in Poisson or TSDF fusion is similar.
> >
> > Indeed, we can reconstruct details at a higher resolution, but enhancing these details does not significantly improve the numerical results. As illustrated in Figure C in our rebuttal PDF, though 2DGS contains more geometric details, it achieves a lower numerical results than our method. Our superiority results come from the way that we estimate a more accurate zero-level set.
> >
> > 3. How we improve accuracy on TNT.
> >
> > As we mentioned in the reply to “2. Visual results.” for reviewer SW8k, we improve the accuracy on TNT by estimating a more accurate zero-level set. Although current methods can reconstruct geometry details, the reconstructions are usually fat, which means their surfaces usually drift away from the real zero-level set. Our method can pull 3D Gaussians onto the zero-level set, which enables us to impose constraints directly on the zero-level set. This really helps us estimate a more accurate zero-level set. Please refer to Figure C in the rebuttal PDF for comparisons of error maps between 2DGS and our method. Additionally, we will provide more quantitative ablation results on TNT dataset in our revision.
> >
> > 6. Constraints on pulled Gaussians.
> >
> > We did not find a proper way to directly implement the constraint of "approximately equal to" in the code.
> > When we were trying verify if “equal to” helps, we tried different weights on this loss term. When the weight is small, we can achieve the same effect as “approximately equal to”, while a large weight can turn the loss into a hard constraint "equal to". But both of them can not work with the uncertainty in the signed distance field during optimization. If you have any better ideas on how to impose a constraint of "approximately zero," please let us know, as we are willing to undertake further attempts.
> >
> > Additionally, our results in Fig.G in our rebuttal PDF were produced with removing far away Gaussians using the same way explained in "5.About far away points".
> >
> > 8. Comparison with GOF.
> >
> > Results in Tab.D are averaged results over indoor scenes, not the averaged results over both indoor and outdoor scenes. The full comparison is provided in the following.
> >
> >
> > |       | **Indoor Scene**    |             |             | | **Outdoor Scene**    |             |             |
> > |-------|---------------------|-------------|-------------|-|----------------------|-------------|-------------|
> > |       | **PSNR$\uparrow$**  | **SSIM$\uparrow$** | **LPIPS$\downarrow$** | | **PSNR$\uparrow$**  | **SSIM$\uparrow$** | **LPIPS$\downarrow$** |
> > | **GOF**   | 30.79              | 0.924       | 0.184       | | 24.82                | 0.750       | 0.202       |
> > | **Ours**  | 30.78              | 0.925       | 0.182       | | 23.76                | 0.703       | 0.278       |
> >
> >
> >
> > As we mentioned in the rebuttal, we can not report the results on all scenes in DTU due to the time limit. We are trying our best to report the results on all scenes before the discussion ends. We are limited to a handful computational resources to finish so many results in the rebuttal PDF.

---

> > > ### Comment · Reviewer_ZhFP · 2024-08-12
> > >
> > > Thanks for your response. I still have the following questions.
> > > **1. Reconstruction details.** I cannot agree that enhancing geometry details does not significantly improve the numerical results. On DTU dataset, many methods have achieved the reconstruction performance of around 0.6, such as Neuralangelo [1]. These method can reconstruct details well, leading to their better performance. In fact, the visualization of the proposed method on TNT dataset is relatively poor, making the good quantitative results a bit unconvincing. As mentioned in your main paper (L249-250), you have split large scale scenes into parts to extract meshes, however, the proposed method cannot still recover details on TNT. In addition, how do you evaluate your reconstructed meshes on TNT dataset?
> > > [1] Li, Zhaoshuo, et al. "Neuralangelo: High-fidelity neural surface reconstruction." Proceedings of the IEEE/CVF Conference on Computer Vision and Pattern Recognition. 2023.
> > > **8. Comparison with GOF.** As shown in the complete Table, in fact, the method degrades the rendering quality a lot. Can you explain it?

---

> ### Author Response · Authors · 2024-08-12
>
> Dear reviewer ZhFP,
>
> Thanks for your questions.
>
> 1. Reconstruction Details
>
> If compared methods can recover comparable surface positions, then more geometry details definitely produce better numerical results. However, if one method fails to recover accurate surface positions, then more geometry details could not improve the numerical results a lot. If a reconstructed surface does not come from an accurate zero-level set, it would have inaccurate surface position which drifts away a lot from the GT surface, which hurts the numerical results.  To show this, we draw a figure below to illustrate the comparison. This figure shows that our surface (▲) is nearer to the ground truth (*), although the geometry details recovered by the marching cubes algorithm are not many. In contrast, the other surface (■) has much more geometry details which also look quite similar to the ground truth, but fails to recover accurate surface position, drifting far away from the ground truth, which hurts the numerical results a lot. This is also indicated quite clearly in Fig.C in our rebuttal PDF, where the error map across the whole surface obtained by 2DGS looks much darker (bigger errors) than ours.
>
> ```plane text
> *: GT surface, ▲: Our surface, ■: 2DGS surface
> ---------------------------------------------------
> Our surface vs. GT
>    *        ***
>   *▲*      *▲  *▲
> ▲▲▲ ▲▲  ▲▲▲▲ ▲▲▲ ▲▲ ▲▲▲▲
>       ▲▲ *        *▲
>        **
> ---------------------------------------------------
> 2DGS surface vs. GT
>    *        ***
>   * *      *   **
> **   *    *      *  ****
>       *  *        **
>        **
>              ■
>             ■ ■
>   ■■■      ■   ■■    ■
> ■■   ■    ■      ■  ■ ■
>       ■  ■        ■■   ■
>        ■■
> ---------------------------------------------------
> ```
>
>
>
> Moreover, we seriously checked our code, especially the evaluation part, we did not see bugs. For fair evaluations on TNT, we use the official github code released with the dataset of the original paper “Tanks and Temples: Benchmarking Large-Scale Scene Reconstruction”.
>
> 8. Comparison with GOF
>
> We achieved comparable rendering quality to GOF in indoor scenes. We also noticed that our rendering quality is worse than GOF in outdoor scenes. The main reason is that outdoor scenes include lots of huge Gaussians, each of which covers a large area. This may be caused by the lack of view coverage. These huge Gaussians usually struggle to meet all the constraints that we imposed for SDF inference. For example, they can not be aligned well with the gradient at the zero-level set, and their large size amplifies little normal adjustment to a large impact on the rendering, since it can not take all area it covers into account. This also raises up a future work direction, that is how to control the size of Gaussians during our inference, so that we can maintain good rendering quality of GS.

---

> > ### Comment · Reviewer_ZhFP · 2024-08-12
> >
> > Thanks for your further response. I still have the following questions.
> > **Reconstruction details.** As you explained, the proposed method cannot recover potential details like 2DGS, although these details are not accurate. The current SOTA methods achieve ~0.6 performance on DTU (lower is better) and ~0.5 performance on TNT dataset (higher is better) as they can recover accurate details. Since Gaussian points are usually dense in details, this should help recover potential details, I wonder what limits the proposed method to recover the potential details.
> > **Comparison with GOF.** Thanks for your explanation. Can you also conduct a rendering ablation study on MipNeRF 360 for the proposed losses like Table B? I wonder what component influences the rendering.

---

> ### Author Response · Authors · 2024-08-12
>
> Dear reviewer ZhFP,
>
> Thanks for your questions.
>
> 1. Reconstruction details
>
> Since our method infers signed distances from 3D Gaussians, we believe the 3D Gaussians are the key to recover geometry details. Although 3D Gaussians are dense (for most time but we still have some sparse cases), they are not dense enough like point cloud scans to recover fine geometry. This is because 3D Gaussian is not a point but a sphere or a plane which covers some space. This is fine for rendering, since the pixel color is usually produced by several overlapped Gaussians, but obviously using one Gaussian is hard to infer fine geometry in the area it covers, especially when the Gaussian is very large, which makes them much different from points back projected from the rendered depth. Just like the extreme case raised up by reviewer SW8k, the large holes in the middle of Figure 15 in the outdoor scene, we observed huge Gaussians in these areas. Although these huge Gaussians may work well in rendering, but they cannot recover any geometry covered by them. Thus, how to control the size of Gaussians for SDF inference could be an interesting future work direction.
>
> Another reason is like what we mentioned in the limitation. The MLP we use to estimate an SDF prefers low-frequency information, which also affects the performance of recovering fine geometry. This can also be resolved in our future work by capturing high-frequency geometry either using high frequency positional encoding or without using MLP.
>
> We will add this discussion in our revision.
>
>
> 2. MipNeRF360 ablation
>
> We conduct additional ablation studies on MipNeRF360 dataset, as shown in the following table. Due to the time limit, we select two scenes from indoor and outdoor scenes, respectively. The terms that are irrelevent to rendering are omiited. We are trying our best to report the results on all scenes along with the ablation results on DTU before the discussion ends. We would like to clarify in advance that ablations related to rendering metrics do not show as clear distinctions as those related to reconstruction metrics. This is because the rendering metrics are usually more stable than reconstruction metrics, which is evident in previous works.
>
> The term "w/o $L_{Tan}$" has the most significant impact on rendering quality because we align the Gaussian normals with the surface normals to obtain consistent orientated Gaussians, which enhances the rendering quality. The term "w/o Pull GS" has the second most significant impact on rendering metrics, because we optimize the Gaussian positions by pulling them towards the zero-level set to achieve better rendering quality. The other terms "w/o $L_{Thin}$", and "w/o $L_{Oth}$" are mainly designed for learning SDF and have a minor improvement on the rendering metrics.
>
> |             | **Indoor**           |           |           | **Outdoor**          |           |           |
> |-------------|----------------------|-----------|-----------|----------------------|-----------|-----------|
> |             | **PSNR↑**            | **SSIM↑** | **LPIPS↓**| **PSNR↑**            | **SSIM↑** | **LPIPS↓**|
> | **w/o Pull GS** | 29.74              | 0.920     | 0.207     | 23.97                | 0.730     | 0.255     |
> | **w/o $L_{Thin}$**  | 30.07              | 0.919     | 0.190     | 24.11                | **0.747**     | **0.237**     |
> | **w/o $L_{Tan}$**   | 29.35              | 0.913     | 0.213     | 22.75                | 0.702     | 0.261     |
> | **w/o $L_{Oth}$**   | 30.18              | 0.927     | 0.190     | 24.11                | **0.747**     | 0.239     |
> | **Full model**  | **30.19**              | **0.929**     | **0.189**     | **24.12**                | **0.747**     | **0.237**     |
>
>
>
> Please feel free to let us know if you have any more questions.
>
>
> Best,
>
> The authors

---

> > ### Comment · Reviewer_ZhFP · 2024-08-13
> >
> > Thank you very much for your response. However, the answer on the **Reconstruction details** cannot address my concern. For many detail regions, the Gaussian points are with small scales. However, the pulling operation cannot reflect these details. As shown in the Figure F, when applying Poisson reconstruction to extract meshes from Guassion points, the dotted details on the front of the Lego can be reconstructed well. This demonstrates the Guassian points can better represent these details. However, the proposed method cannot reconstruct these details. In addition, the high frequency positional encoding and the Instant-NGP are commonly used with MLP to reconstrcut surfaces. I wonder why the proposed method does not leverage these techniques. Can you combine these tehniques with the proposed method to test the Ignatius scene of the TNT dataset and show the evaluation results?

---

> > > ### Author Response · Authors · 2024-08-13
> > >
> > > Thanks for your comments. We are pleased to discuss more about the details.
> > >
> > > 1. Ours (Screened Poisson) in Fig. F in our rebuttal PDF was obtained using points back projected from rendered depth rather than 3D Gaussians.
> > >
> > > Reviewer ZhFP misunderstood how our mesh reconstruction was produced by the screen Poisson. As we mentioned in our response “1. Reconstruction details” posted at 14:53 on Aug.7, we used points back projected from rendered multi-view depth maps to reconstruct the mesh surface with Poisson, rather than the raw Gaussian points.
> > >
> > > Points from depth maps have been used by SuGaR and GaussianSurfels to reconstruct surfaces with Poisson reconstruction, and we have not seen methods directly using Gaussian positions to reconstruct surfaces so far. The reasons are twofold. One is that Gaussians are not guaranteed to be dense everywhere in a scene since they are just responsible for good rendering quality, some poor reconstruction caused by sparse Gaussians can be found in the large holes in the middle of Fig.15 in the outdoor scene in our paper. Another key reason, we believe, is that Gaussian positions do not accurately represent the surface, even if previous methods and we have tried various constraints, such as our pulling, to locate Gaussian positions on the estimated surface. However, splatting makes a difference. When we splat these Gaussian on a 2D plane and render them, the Gaussians in the view frustum will work together to approximate more accurate geometry and also surface details inferred from multi-view photometric consistency, just like what we show in Fig. F in our rebuttal PDF. Thus, Gaussian positions do not represent geometry details quite well, but depth maps rendered by splatting Gaussians do.
> > >
> > > We believe pulling works well in recovering geometry details, since it has been justified by high fidelity reconstructions in NeuralPull[1], and its following works, such as [2-4].
> > >
> > > 2. Why not Instant-NGP
> > >
> > > As we explained in our previous post, using Instant-NGP like hash encoding and high-frequency positional encoding will be our future work. We did not use them in this project, since we have seen pretty stunning reconstructions obtained by previous methods using pulling with MLP. Although most of these works are working with point clouds (not learnable), we thought this mature framework can also recover geometry details from 3D Gaussians (learnable) with proper other designs. Thus, this is a straightforward and intuitive attempt, but definitely a good start for us to keep working in this direction.
> > >
> > > We will start working with integrating hash encoding with the SDF representation in our current method immediately, and try our best to post results as you requested before the reviewer-author discussion period ends. Our implementation may take some time. If we are unable to complete it before the discussion period ends, we guarantee that we will further discuss this point in the revision.
> > >
> > > [1]. Ma B, Han Z, Liu Y S, et al. Neural-Pull: Learning Signed Distance Function from Point clouds by Learning to Pull Space onto Surface. International Conference on Machine Learning. PMLR, 2021: 7246-7257.
> > >
> > > [2]. Ma B, Liu Y S, Zwicker M, et al. Surface reconstruction from point clouds by learning predictive context priors. Proceedings of the IEEE/CVF Conference on Computer Vision and Pattern Recognition. 2022: 6326-6337.
> > >
> > > [3]. Ouasfi A, Boukhayma A. Few-Shot Unsupervised Implicit Neural Shape Representation Learning with Spatial Adversaries. Forty-first International Conference on Machine Learning.
> > >
> > > [4]. Chou G, Chugunov I, Heide F. Gensdf: Two-stage learning of generalizable signed distance functions. Advances in Neural Information Processing Systems, 2022, 35: 24905-24919.

---

> > > > ### Author Response · Authors · 2024-08-13
> > > >
> > > > 3. More thorough ablation studies
> > > >
> > > > We have completed the ablation studies for all scenes on DTU and MipNeRF360. The results are listed in the tables below. We will add these results in our revision. The conclusions drawn from this version of the ablation study, which is conducted on all scenes, are largely in agreement with the results from the previous version of the ablation study, which was conducted on a subset of scenes.
> > > >
> > > > **Ablation studies on DTU dataset across all scenes**
> > > >
> > > > | | Pulling            |               | Constraints            |               |               | Mesh  |      |            |
> > > > |---------|--------------------|---------------|---------------|---------------|---------------|------------------|------|------------|
> > > > |         | **Pulled to centers**  | **w/o Pull GS**   | **w/o $L_{Thin}$** | **w/o $L_{Tan}$** | **w/o $L_{Oth}$** | **TSDF**             | **Poisson** | **Full model** |
> > > > | **CD↓**     | 0.90               | 0.85          | 0.78          | 0.83          | 0.79          | 1.41             | 0.79    | **0.74**       |
> > > >
> > > >
> > > > **Ablation studies on MipNeRF360 dataset across all scenes**
> > > >
> > > > |            | **Indoor**       |           |            | **Outdoor**      |           |            |
> > > > |------------|------------------|-----------|------------|------------------|-----------|------------|
> > > > |            | **PSNR↑**        | **SSIM↑** | **LPIPS↓** | **PSNR↑**        | **SSIM↑** | **LPIPS↓** |
> > > > | **w/o Pull GS**  | 29.97            | 0.898     | 0.191      | 22.47            | 0.684     | 0.293      |
> > > > | **w/o $L_{Thin}$**| 30.57         | 0.922     | 0.185      | 23.63            | 0.699     | 0.279      |
> > > > | **w/o $L_{Tan}$** | 29.65         | 0.897     | 0.207      | 22.05            | 0.674     | 0.300      |
> > > > | **w/o $L_{Oth}$** | 30.69         | 0.923     | **0.182**      | 23.74            | 0.702     | **0.278**      |
> > > > | **Full model**  | **30.78**           | **0.925**     | **0.182**      | **23.76**            | **0.703**     | **0.278**      |

---

> > > > > ### Comment · Reviewer_ZhFP · 2024-08-14
> > > > >
> > > > > Thank you very much for your comprehensive ablation studies. Since I am curious about the potential of the method on reconstruction details, I am looking forward to your experimental results on the reconstruction details using Instant-NGP or high-frequency positional encoding.

---

> ### Author Response · Authors · 2024-08-14
>
> We tried two implementations to add high-frequency information. The first one is appending positional encoding to the input XYZ coordinates, like the original NeRF. The other is replacing the input 3D points with multi-resolution hash encoding, like Instant-NGP. All parameter settings are consistent with those of the original implementations. The experiment is conducted on the Ignatius scene of the TNT dataset as requested. Using the same experimental setting, as reported in the following table, both adding positional encoding and hash feature grid from instant-ngp can slightly improve the reconstruction visually and numerically. We do not have time to try more parameter options like frequencies in pe and grid resolution in hash feature grid, but we will report that in our revision.
>
> Ours: 0.71
>
> +positional encoding: 0.73
>
> +hash encoding: 0.75

---

> > ### Comment · Reviewer_ZhFP · 2024-08-14
> >
> > Thank you very much for your feedback. I have increased my score!

---

### Official Review · Reviewer_SW8k · 2024-07-12

**Soundness:** 1
**Presentation:** 3
**Contribution:** 2
**Rating:** 5
**Confidence:** 5

**Summary:**

This paper combines 3D Gaussian Splatting and NeuralPull to extract surface. By this way, it can utilize the existing extracting method Marching Cubes algorithm to extract the zero-level set as a mesh surface.

**Strengths:**

1. With a neural SDF network, this paper can utilize the Marching Cubes algorithm to extract the surface instead of TSDF fusion or Poisson.
2. The paper uses three datasets to validate the proposed method.

**Weaknesses:**

1. Lack of explanation of how this method can work. As we all know, NeuralPull needs point clouds as ground truth to let a neural SDF learn surface, while the pseudo ground truth provided by 3D Gaussians is noisy, so how can your method learn? In addition, based on your description, you jointly train the 3D Gaussian and the neural SDF and use Eq.2 to pull the 3D Gaussian onto the zero-level set. However, at the beginning of training, the neural SDF will not provide good guidance and even provide a poor direction leading to making the 3D Gaussian away from the surface, leading to catastrophe.

2. Unsatisfactory visual results on TNT. Although your F1-score is better than 2DGS, in Fig.14 your results look over smooth and lack details. This figure only shows your results, can you provide more comparisons with other methods, like 2DGS? Also, in Fig.15, why are there many holes in the mesh in the middle?

3. The results of 2DGS are worse than the original paper. For example, on the TNT dataset, 2DGS is 0.3 in the original paper, while it is 0.26 in Fig.2 of your paper. Also, the qualitative results of 2DGS are better than yours. In detail, the mesh shown in Fig.10 of 2DGS's paper has more details, which are better than yours, and 2DGS's results are shown in Fig.4 of your paper.

4. Your quantitive results are not consistent between Tab.2 and Tab.4 on TNT. In detail, in Tab.2, the full model F1-score is 0.43, while in Tab.4 it is 0.46, leading to suspicion of the accuracy of your results. Can you explain it?

5. Lack of correct citation. For example, your first constraint $L_{Thin}$ is actual $L_s$ in NeuSG, but you don't illustrate the origin.

**Questions:**

Some questions have been listed in the weaknesses. I still have other questions here:
1. According to [1,2], SDF with Neural Implicit Functions (NIFs) can only reconstruct closed surfaces. The limitation prevents NIFs from representing most real-world objects. However, on the TNT dataset, some scenes are not closed surfaces, like 'Barn' with the ground, 'Meetingroom' with windows. How do you solve this problem using SDF with NIFs?

2. For Eq.4 and Eq.6, you use L1 loss to regulate vectors, however, it is more common to use the cosine distance function to regulate vectors. Can you explain why you use L1 loss instead of cosine distance?

3. For rendering, you only use L2 loss, however, D-SSIM is also used to optimize Gaussian splatting. Can you explain why you don't use it in your method?

4. The method is based on NeuralPull, which learns a neural SDF from ground truth point clouds, while this method optimizes a neural SDF from 3D Gaussian splatting. Therefore, the original NeuralPull is the upbound of this method. Can you provide an experiment that trains NeuralPull on the TNT dataset and compares it with it to show the upbound of the proposed method?

5. How do you deal with the surface of 'ground'? For example, in Fig.4 (right, 'Truck') and Fig.14 (left, 'Barn'), the ground is built. However, in Fig.14 (middle, 'Caterpillar'), the ground is missing. Do you manually delete it or something else? In your evaluation, do you also delete it? As I know, on the TNT dataset, the extracted meshes are not modified manually for evaluation.


[1] CAP-UDF: Learning Unsigned Distance Functions Progressively from Raw Point Clouds with Consistency-Aware Field Optimization.

[2] Neural unsigned distance fields for implicit function learning.

**Limitations:**

In Fig.12, the paper shows this method can not reconstruct the details of 'lego'. Can the authors show the comparsion with this method and SuGar as well as 2DGS?

---

> ### Author Rebuttal · Authors · 2024-08-07
>
> **1. Explanation of the method.** When training 3DGS, 3D Gaussians are progressively approaching to the zero-level set. At the same time, we pull Gaussians to align with the zero-level set of SDF. Under the joint optimization of these two process, along with our novel pulling operation and constraints, the Gaussian point clouds that get pulled on the zero-level set gradually turn out to be clean, which can be used as the pulling target to infer signed distance in the field, leading to surfaces fitted by the SDF that gradually recovers detailed and accurate geometry.
>
> We agree that the learned SDF field is coarse at the very beginning, which is not a good guidance for Gaussians. Therefore, we firstly train 3DGS for 7000 epochs, leading to relatively stable Gaussians, and then start to pull queries onto Gaussians to estimate a rough SDF, and finally we pull Gaussians on to the zero-level set, and queries are also following to get pulled onto the pulled Gaussians since then.
>
> **2. Visual results.** The surfaces learned by 2DGS are usually fat and a little bit drift away from ground truth surfaces, although their meshes seem to show more details. Our method is able to capture more accurate surfaces by using 3D Gaussians pulled onto the zero-level set and pulling query points onto Gaussian disks at the same time, leading to much more accurate zero-level set. We additionally report comparisons of error maps on meshes obtained by 2DGS and ours in **Fig. C** in rebuttal PDF, which highlights our superiority in terms of the accuracy of extracted surfaces. Additionally, the holes in the middle of Figure 15 are due to the ill-conditioned distribution of Gaussians, making it difficult to distinguish geometric structures.
>
> **3. 2DGS numerical results.** 2DGS wrongly reported the mean result on TNT in its original paper. The calculated average value is indeed 0.26.
>
> **4. Results between Tab.2 and Tab.4.** Thanks for pointing this out. Before the deadline, we updated our results frequently, but we forgot to update the latest results everywhere in the paper. We wrongly reported an older version of ablation study, which was conducted on the first four scenes in TNT dataset. We will update the ablation study with the final version, which was conducted on all of the TNT scenes, as reported in **Tab. B** in the rebuttal PDF.
>
> **5. $L_{Thin}$.** We did mention NeuSG in Line 175. We'll make it more clear that the thin loss is inspired by NeuSG in our revision.
>
> **6. Closed Surfaces.** SDF can also represent open structures. As a result, it will reconstruct double-layer surfaces. You can refer to the second and third row of “NeuS” in Figure 4 in the paper of NeuralUDF (CVPR2023), where the SDFs successfully distinguish the collar and cuffs, but the reconstructed cloths have tight double-layer surfaces. To avoid the influence of double-layer surfaces on evaluation accuracy, we practically delete the back faces according to the visibility of each face under each camera view. Through this way, we can accurately reconstruct open structures with single-layer surfaces. Additionally, although UDF can reconstruct open surfaces, extracting the zero-level set from UDF as a mesh surface is still a challenge, resulting in artifacts and outliers on the reconstructed meshes. Our method can also learn UDF, and we additionally conduct an experiment using the same setting to learn UDF and compare surfaces extracted from the SDF and the UDF, as shown in **Fig. D** in the rebuttal PDF, which is a corner of “Barn”, showing the shortcomings of UDF learning.
>
> **7. Normal loss.** We also tried cosine similarity as the normal loss. We did not see any difference on the performance for these two constraints on normal.
>
> **8. D-SSIM loss.** We also tried the D-SSIM loss. But we did not see any difference on the performance. For simplicity, we do not include it in the loss function.
>
> **9. Performance of NeuralPull.** In this paper, we resolve the problem of learning SDF from multi-view images through sparse and noisy Gaussians by innovatively pulling queries onto Gaussian disks on the zero-level set of the SDF. Although we are using pulling, our pulling is much different from NeuralPull. The difference lies in that NeuralPull pulls a query to a point while we pull a query onto a Gaussian disk. 3D Gaussians are constrained to be a disk-like shape which covers an area rather than a point during optimization, which inspires us to introduce this pulling variant. Therefore, the performance of our method is not limited to the bottleneck of NeuralPull. Figure 2 in our paper shows that merely pulling queries to the point cloud, like NeuralPull, does not work well with 3D Gaussians, due to the sparsity. Additionally, we conduct an experiment that using a sparse ground truth point cloud of “Ignatius'” scene to train both NeuralPull and our pulling operation. The point clouds are initialized as spheres to simulate the Gaussians used by our method. The results in **Fig. E** in our rebuttal PDF highlights the superiority of our proposed method over NeuralPull.
>
> **10. Ground of surface.** The ground truths of TNT dataset have provided bounding box for culling the reconstructed meshes. For example, the ground truth points of “Truck” and “Barn” contain the ground while the points of “Caterpillar” do not contain the ground. We first crop the meshes using the provided bounding box and then visualize them.
>
> **11. Reconstructing details.** Since we use the marching cubes algorithm to extract the zero-level set as a mesh surface, our results is limited by the resolution used by the marching cubes. While 2DGS uses multi-view depth images to reconstruct meshes with Poisson surface reconstruction, which achieves higher resolution and recover more geometry details. We additionally use the same way of reconstruction as 2DGS, and show visual comparison in **Fig. F** in our rebuttal PDF, where we can see that our reconstruction show more geometry details than 2DGS.

---

> > ### Comment · Reviewer_SW8k · 2024-08-11
> >
> > Thank you for your rebuttal. I still have some questions:
> > 1. Why do you say '2DGS wrongly reported the mean result on TNT in its original paper'? Can you explain it or give some proof?
> > 2. About 'Performance of NeuralPull': The Fig. E in your rebuttal PDF only shows one scene. Can you give a table across six scenes from TNT dataset to show your method is better? Also, the NeuralPull used in your experiment is trained on sparse point clouds from colmap, right? Since TNT has the ground truth point cloud, can you use the ground truth point cloud to train the NeuralPull. I just want to see the upbound of your method, which is the NeuralPull trained with ground truth. It's okay that the neuralpull trained with ground truth is better than your method.
> > 3. About 'the holes in the middle of Figure 15 are due to the ill-conditioned distribution of Gaussians'. However, why is there not a hole in the right figure in Fig. 15 of your paper? What's the difference between the middle and the right?
> > 4. About 'UDF': The Fig. D in your rebuttal PDF only shows one scene. Can you give a table across six scenes from TNT dataset to show your method is better?

---

> > > ### Author Response · Authors · 2024-08-11
> > >
> > > Dear reviewer SW8k, Thanks for your comments. We
> > > are happy to answer your questions.
> > >
> > > 1. About 2DGS numerical results.
> > >
> > > According to Table 2 in 2DGS paper, if we average the per-scene result provided in the table, we can get (0.36+0.23+0.13+0.44+0.16+0.26)/6=0.26, but 2DGS reported the mean value as 0.30 instead of 0.26. Thus we say '2DGS wrongly reported the mean result on TNT in its original paper'.
> > >
> > > 2. About "Performance of NeuralPull".
> > >
> > > We trained both NeuralPull and our method on sparse *ground truth* point clouds, instead of COLMAP point clouds in our rebuttal (see our response "9. Performance of NeuralPull"). Following your advice, we additionally conduct experiments on all of the six scenes of TNT dataset, following the same setting used in our response "9. Performance of NeuralPull." The numerical results in terms of F-score are reported in the following table.
> > >
> > > | Methods    | Barn | Caterpillar | Courthouse | Ignatius | Meetingroom | Truck | Mean |
> > > |------------|------|-------------|------------|----------|-------------|-------|------|
> > > | NeuralPull | 0.70 | 0.40        | 0.51       | 0.51     | 0.56        | 0.73  | 0.57 |
> > > | Ours       | **0.78** | **0.58**   | **0.66**   | **0.66** | **0.65**    | **0.76**  | **0.68** |
> > >
> > > The comparison indicates that our method is a variant of NeuralPull, but not exactly the same as NeuralPull. Thus, we can produce much better accuracy on points than the upper bound of NeuralPull, due to the ability of pulling queries on plans rather than points.
> > >
> > > 3. About the holes.
> > >
> > > Due to overly complex geometric structures and a lack of view covering, there is a significant under-fitting issue in the flowerbed area of the middle scene. This results in a set of extremely sparse, huge, and unevenly distributed Gaussians, which makes Gaussians are thick ellipsoid like shape rather than relatively thin plans, leading to poor sense of surface. While this issue does not occur in other scenes.
> > >
> > > 4. About learning UDF.
> > >
> > > Following your advice, we also additionally conduct experiments of learning UDFs on all of the six scenes in TNT dataset, as reported in the following table, in terms of F-score. The superiority results of learning SDF beyond learning UDF demonstrates the drawbacks of UDF.
> > >
> > > | Methods    | Barn | Caterpillar | Courthouse | Ignatius | Meetingroom | Truck | Mean |
> > > |------------|------|-------------|------------|----------|-------------|-------|------|
> > > | NeuralPull | 0.55 | 0.34        | 0.15       | 0.60     | 0.17        | 0.46  | 0.38 |
> > > | Ours       | **0.60** | **0.37**   | **0.16**   | **0.71** | **0.19**    | **0.52**  | **0.43** |
> > >
> > > The comparison in the above table indicates that our method can learn not only SDF but also UDF, which also show dvantages over NeuralPull in learning of UDFs.

---

> > > > ### Comment · Reviewer_SW8k · 2024-08-12
> > > >
> > > > Your answers solve most of my concerns. I'm happy to raise my rating. However, I still have some questions here.
> > > > 1. About "Performance of NeuralPull": How do you train both your method on sparse ground truth point clouds, instead of COLMAP point clouds? I think in your paper, you train your method on sparse ground truth point clouds from colmap, right?
> > > > 2. I'm still confused that your f1 score is not consistent with your visualization on TNT dataset. I think 0.43 F1 score is high, but your visualization on TNT dataset is really poor.

---

> > > > > ### Author Response · Authors · 2024-08-12
> > > > >
> > > > > Dear reviewer SW8k,
> > > > >
> > > > > Thanks for raising your rating. We are glad to discuss more details here.
> > > > >
> > > > > 1. About "Performance of NeuralPull".
> > > > >
> > > > > In our rebuttal, in order to intuitively compare the pure pulling operation between NeuralPull and our method, we use a set of sparse ground truth point clouds, which were downsampled at a factor of 2000 from the ground truth points provided by TNT dataset to conduct the experiments. We did not use the sparse point clouds estimated by COLMAP, because we want to exclude the potential misguidance of the inaccurate points. We initialize the points as flat spheres to simulate the 3D Gaussians, and fix the points during training. The parameters to be optimized are only the SDF network parameters. This experiment highlights the differences between NeuralPull and our proposed method.
> > > > >
> > > > > Moreover, in the main paper, the initial point clouds are from COLMAP, which initialize the learnable Gaussians and is the basic procedure in 3DGS. After that, we do not use the initial point cloud any more. Then, we learn Gaussians through splatting, simultaneously the positions and the shapes of Gaussians which are used to estimate the SDF. Obviously, this is a dynamic optimization procedure, both Gaussians and SDF are updated iteration by iteration, which is not like what NeuralPull can do using a fixed set of points as pulling targets. Here, we got a coarse SDF by pulling randomly sampled queries to their nearest Gaussian disks on the zero-level set, meanwhile, we also pull Gaussians onto the zero-level set, and directly impose constraints on these pulled Gaussians, including the normal alignment, minimal rendering errors, and flat plans, etc.. Thus, we use pulled Gaussians (not fixed) as the pulling targets. Moreover, with these pulling targets, queries are pulled onto the Gaussian disks rather than the centers, as shown in Eq.5 in our paper, which aims to handle the sparsity of Gaussians and also differentiates our method from the original NeuralPull.
> > > > >
> > > > > 2. About the results
> > > > >
> > > > > As we explained in our response “2. Visual results. ” in the rebuttal above, their numerical results are not as good as their visual results, since the reconstructed surfaces do not come from an accurate zero-level set. To show this, we draw a figure below to illustrate the comparison.
> > > > >
> > > > > This figure shows that our surface (▲) is nearer to the ground truth (*), although the geometry details recovered by the marching cubes algorithm are not many. In contrast, the other surface (■) has much more geometry details which also look quite similar to the ground truth, but fails to recover accurate surface position, drifting far away from the ground truth, which hurts the numerical results a lot. This is also indicated quite clearly in Fig.C in our rebuttal PDF, where the error map across the whole surface obtained by 2DGS looks much darker (bigger errors) than ours.
> > > > >
> > > > > ```plane text
> > > > > *: GT surface, ▲: Our surface, ■: 2DGS surface
> > > > > ---------------------------------------------------
> > > > > Our surface vs. GT
> > > > >    *        ***
> > > > >   *▲*      *▲  *▲
> > > > > ▲▲▲ ▲▲  ▲▲▲▲ ▲▲▲ ▲▲ ▲▲▲▲
> > > > >       ▲▲ *        *▲
> > > > >        **
> > > > > ---------------------------------------------------
> > > > > 2DGS surface vs. GT
> > > > >    *        ***
> > > > >   * *      *   **
> > > > > **   *    *      *  ****
> > > > >       *  *        **
> > > > >        **
> > > > >              ■
> > > > >             ■ ■
> > > > >   ■■■      ■   ■■    ■
> > > > > ■■   ■    ■      ■  ■ ■
> > > > >       ■  ■        ■■   ■
> > > > >        ■■
> > > > > ---------------------------------------------------
> > > > > ```

---

> > > > > > ### Comment · Reviewer_SW8k · 2024-08-12
> > > > > >
> > > > > > Thank you for your response. I'm willing to raise my rating to borderline accept.

---

> > > > > > > ### Author Response · Authors · 2024-08-12
> > > > > > >
> > > > > > > Dear reviewer SW8k,
> > > > > > >
> > > > > > > Thanks a lot for your comments and suggestions. We sincerely appreciate your accept recommendation.
> > > > > > >
> > > > > > > Best,
> > > > > > >
> > > > > > > The authors

---

> > > > > > > > ### Comment · Reviewer_SW8k · 2024-08-13
> > > > > > > >
> > > > > > > > Will you release your code in the future?

---

> > > > > > > > > ### Author Response · Authors · 2024-08-13
> > > > > > > > >
> > > > > > > > > Yes, we will release the code upon paper acceptance.

---

### Official Review · Reviewer_5rKf · 2024-07-12

**Soundness:** 3
**Presentation:** 3
**Contribution:** 3
**Rating:** 7
**Confidence:** 5

**Summary:**

This paper focuses on the challenge of inferring a signed distance function (SDF) for multi-view surface reconstruction from 3D Gaussian splatting (3DGS), which is hindered by the discreteness, sparseness, and off-surface drift of the 3D Gaussian. To overcome these challenges, the authors propose a method that seamlessly integrates 3DGS with the learning of neural SDFs. This approach constrains SDF inference with multi-view consistency by dynamically aligning 3D Gaussians on the zero-level set of the neural SDF and then rendering the aligned 3D Gaussians through differentiable rasterization. Through the utilization of both differentiable pulling and splatting, the approach jointly optimizes 3D Gaussians and neural SDFs with both RGB and geometry constraints, resulting in the generation of more accurate, smooth, and complete surfaces. Extensive experimental comparisons on various datasets demonstrate the superiority of the proposed method.

**Strengths:**

- The approach of optimizing the neural Signed Distance Function (SDF) using only the regularization from 3DGS, without any monocular geometry supervision, is both novel and compelling. It effectively circumvents the time-consuming point sampling process for volume rendering, commonly utilized in prior studies for SDF learning. The method's natural balance between 3DGS and the neural SDF is simple and intuitive, showcasing the potential for seamless integration of explicit 3DGS and implicit neural fields.

- The proposed method has demonstrated state-of-the-art reconstruction results on the DTU and TNT datasets, with each component of the approach undergoing thorough examination in ablation studies.

- This paper is well-structured and easy to follow.

**Weaknesses:**

- It seems that the authors did not incorporate the Eikonal loss, which is commonly used to regularize SDF values in space during optimization. Could the authors provide specific reasons for this omission? Additionally, it is not entirely clear to me how the pull loss ensures that the optimized field satisfies the properties of SDF.

- In Eq8, the weight of the splatting loss appears to be substantially larger than the others. Could the authors explain how these weights are determined?

- I noticed that the densification operation and the pull/constraint losses are employed at different stages. How does the simultaneous use of densification and pull loss affect the final result? Stopping the densification operation in the second stage seems unreasonable because it may limit the performance of 3DGS.

**Questions:**

See weakness for details.

**Limitations:**

The authors have thoroughly discussed the limitations of the proposed method. I believe that incorporating monocular priors during optimization would be beneficial to the learning of SDF. Additionally, I think that further exploration of the relationship between densification and SDF values would be an interesting and valuable research pursuit.

---

> ### Author Rebuttal · Authors · 2024-08-07
>
> **1. Eikonal loss.** We believe pulling based methods may not need an Eikonal loss to guarantee the learned distance field is an SDF. This is because we use a normalized gradient and a predicted signed distance when pulling a query. It has been proved by NeuralPull[1] that adding the Eikonal loss will significantly degenerate the learning of SDF, as reported by ``Gradient constraint'' in Table 8 in the original paper of NeuralPull. The reason is that NeuralPull depends on both predicted SDF values and gradient directions to optimize the SDF field. It makes the optimization even more complex when adding additional constraint on the gradient length. We also additionally report a result with an Eikonal loss in **Fig. B** in our rebuttal PDF. The comparison indicates that the results degenerate significantly with an Eikonal loss.
>
> **2. SDF property.** It has been proved that an MLP which is trained using a pulling loss can converge to a signed distance function. The reason is that the sign of distances needs to turn over when across the surface points so that the pulling loss can be minimized on both sides of the surface points. Please see Theorem 1 in the original paper of NeuralPull for more details.
>
> **3. Weight of splatting loss.** We set the weight of splatting loss to 1.0, following all 3DGS baseline methods. And we set the weights of thin loss, tangent loss, pull loss, orthogonal loss as 100, 0.1, 1, 0.1, respectively. The weight of thin loss is larger than others because the scaling factor is usually small and we want the smallest scaling factor to be as small as possible, which is the same as NeuSG[2]. A smaller weight of thin loss will cause Gaussians to be not flat enough, leading to an inaccurate alignment of Gaussians to the zero-level set.
>
> **4. Densification operation and pull loss.** There are three reasons why we separate the densification operation and pulling operation. **(1)** The 3D Gaussians are sparse and unstable at the early stage of training. Current works usually start to add additional losses after certain epochs of training 3DGS, such as the distortion loss (Eq.13) in 2DGS[3] and the regularization loss (Eq.8) in SuGaR[4]. **(2)** The densification operation causes the number of Gaussians to grow rapidly, leading to a collapse of memory overflow. Therefore, current works generally stop densification after certain epochs of training. For example, original 3DGS stops densification at 15000 epoch while SuGaR stops it at 7000 epoch. Here we follow the setting of SuGaR. **(3)** The pulling operation needs to sample query points for each Gaussian, which is time-consuming. Therefore, we only sample query points for all Gaussians once before adding pulling loss for efficiency. If the number of Gaussians are changing during densification, we need to re-sample query points, which will bring additional time consumption. We conduct an ablation study on different stages of stopping densification or start pulling, on DTU scan37 scene, as reported in **Tab. A** in our rebuttal PDF. The first one is to start pulling loss from beginning, which significantly degenerates the performance and slows down the convergence. Another one is to add densification operation until end of training, which takes much longer training time and degenerates the performance.
>
> [1]. Ma B, Han Z, Liu Y S, et al. Neural-Pull: Learning Signed Distance Function from Point clouds by Learning to Pull Space onto Surface. International Conference on Machine Learning. PMLR, 2021: 7246-7257.
>
> [2]. Chen H, Li C, Lee G H. NeuSG: Neural Implicit Surface Reconstruction with 3D Gaussian Splatting Guidance. arXiv preprint arXiv:2312.00846, 2023.
>
> [3]. Huang B, Yu Z, Chen A, et al. 2D Gaussian Splatting for Geometrically Accurate Radiance Fields. ACM SIGGRAPH 2024 Conference Papers. 2024: 1-11.
>
> [4]. Guédon A, Lepetit V. SuGaR: Surface-Aligned Gaussian Splatting for Efficient 3D Mesh Reconstruction and High-Quality Mesh Rendering. Proceedings of the IEEE/CVF Conference on Computer Vision and Pattern Recognition. 2024: 5354-5363.

---

> > ### Comment · Reviewer_5rKf · 2024-08-13
> >
> > Thanks for the efforts of the authors. Most of my concerns have been addressed. I am willing to improve my rating.

---

> > > ### Author Response · Authors · 2024-08-13
> > > **Thanks for the accept recommendation**
> > >
> > > Dear reviewer 5rKf,
> > >
> > > Thanks for your time and expertise. We are glad to know that our rebuttal addressed your concerns and you are willing to raise your rating. We really appreciate the accept recommendation.
> > >
> > > We will also follow your advice to revise our paper accordingly.
> > >
> > > Best,
> > >
> > > The authors

---

### Official Review · Reviewer_CCgx · 2024-07-13

**Soundness:** 3
**Presentation:** 4
**Contribution:** 3
**Rating:** 7
**Confidence:** 4

**Summary:**

The paper proposes a extension to 3D Gaussian Splatting by making it consistent with a neural SDF that is learnt along with the 3D Gaussians. To make it consistent, the Gaussians are projected to the zero level set, and the neural SDF is optimized to represent the SDF of the surface implied by the Gaussians. As the latter requires the Gaussians to be thin (essentially discs), they encourage this with a loss, and further encourage other properties that hold when the two representations are consistent.

**Strengths:**

- Simple but effective method with well thought out losses
- Good visualisations to explain the benefits of components (Figs 2,6-9)

**Weaknesses:**

- The proposed method uses a loss to make the Gaussians to be flat, why not directly use 2D Gaussians/Surfels? Those papers mention that direct parameterisation is much better than using a loss to flatten Gaussians, why does your method not suffer from issues from not having an exactly flat Gaussian? It would be interesting to have a comparison of your method with 2D Gaussians. However, I would expect at least some discussion of this (whether it is actually important to have exactly flat Gaussians or not).
- While I like this way of constraining the Gaussians better than previous methods, the novelty is a bit limited since there are similar methods (improving GS using SDFs and having Gaussians be flat)

**Questions:**

- For $L_{Tangent}$, how do you determine orientation of the normal?
- Ablations study results don't line up with each other and with TNT results (0.41 vs 0.42 vs 0.43 vs 0.46)? A little confused/concerned here.
- End of page 5, should $n_j$ be $\bar{n}_j$?

**Limitations:**

Addressed in the appendix/supplemental

---

> ### Author Rebuttal · Authors · 2024-08-07
>
> **1. Comparison between flat loss and surfels.** Our method requires calculating the inverse of a three-dimensional covariance matrix (Eq.(5) in our paper) to deterimine the distribution probability of a query point within its nearest Gaussian ellipsoid. This allows us to maximize the probability for pulling query points towards nearest Gaussians. The surfel setting, however, merely provides two dimensional of scaling factors, which does not meet our requirement. In general, pushing Gaussians to be flat or setting Gaussians as surfels serves a similar purpose in mitigating the bias of rendering depth and determining Gaussian normals, thus the flatting loss is a reasonable choice. As an evidence, we replace our flat loss (Eq.(3) in our paper) with the surfel setting in GaussianSurfels[1], as shown in **Fig. A** in the rebuttal PDF. The optimization procedure fails at the start with surfel setting, which demonstrates that it is not a good choice in the differentiable pulling operation.
>
> **2. About L_tangent.** We consider the direction of the smallest scaling factor as Gaussian normals, following previous works[2,3]. The loss pushes the normal to get aligned with the gradient on the zero-leve set.
>
> **3. About $n_j$.** It should be $\bar{n_j}$. We will correct it in our version.
>
>
>
> [1]. Dai P, Xu J, Xie W, et al. High-quality Surface Reconstruction using Gaussian Surfels. ACM SIGGRAPH 2024 Conference Papers. 2024: 1-11.
>
> [2]. Guédon A, Lepetit V. SuGaR: Surface-Aligned Gaussian Splatting for Efficient 3D Mesh Reconstruction and High-Quality Mesh Rendering. Proceedings of the IEEE/CVF Conference on Computer Vision and Pattern Recognition. 2024: 5354-5363.
>
> [3]. Chen H, Li C, Lee G H. NeuSG: Neural Implicit Surface Reconstruction with 3D Gaussian Splatting Guidance. arXiv preprint arXiv:2312.00846, 2023.

---

> > ### Comment · Reviewer_CCgx · 2024-08-11
> >
> > My weaknesses/questions have been sufficiently addressed. However, looking at the other reviewer's comments and the authors rebuttal to them, I have the following concerns:
> > - I was not aware of NeusG, and after looking into it, their method seems highly related. NeusG is essentially GS losses + IGR losses + thin loss + consistency losses, while the proposed method is GS losses + NP losses + thin loss + consistency losses. However NeusG is barely mentioned: the authors respond that it is mentioned in line 175 but that is a citation about the general trend of flattening Gaussians. There should be at least a sentence only about NeusG given it is very similar. Either the work is considered concurrent work and should be stated as such, or it is considered a work that inspired your work (including having first introduced the thin loss which you borrow from them) and this should be made clear in the related work. Furthermore, for the latter case NeusG should also compared to in the results!! Their main results are on TnT and it looks like your method performs better, but can you check they use the same settings and then put it into your paper?
> > - As for the other comments, while some seem quite concerning, the authors have addressed most of them sufficiently. The things that are left are:
> > 	- Normal loss: since cosine sim is what NeusG does and you change it, there should be an ablation on it (in the final version of the paper, not suggesting you get results in time to show us)
> > 	- '2DGS wrongly reported the mean result on TNT in its original paper': also want explanation about this statement, did you consult the authors and figure this out, or are you saying there is a numerical error in the mean operation (doesn't seem to be?)

---

> > > ### Author Response · Authors · 2024-08-11
> > >
> > > Dear reviewer CCgx,
> > >
> > > Thanks for the comments. It is a pleasure to further clarify some points.
> > >
> > > **1. We are much different from NeuSG**
> > >
> > > *i) The task*
> > >
> > > Our method directly learns to capture the geometry inherently represented by 3D Gaussians, while NeuSG aims at utilizing Gaussians as guidance to improve the effect of neural implicit reconstruction.
> > >
> > >
> > > *ii) The framework*
> > >
> > > Our method is based on 3DGS (learning radiance fields from multi-view images) and NeuralPull (learning SDF from points), and we train SDFs by unsupervisedly pulling query points onto the Gaussian disk. In contrast, NeuSG is based on 3DGS (learning radiance fields from multi-view images) and NeuS (learning NeRF and SDF from multi-view images), therefore it trains SDFs through classical neural rendering techniques, including ray marching, point sampling and volume rendering. This makes the whole framework so different from each other.
> > >
> > > *iii) Losses*
> > >
> > > We would like to clarify that, we introduce a variant of NeuralPull, which pulls queries on Gaussian plans but rather on points like the original NeuralPull. To achieve this, our pulling loss is different from the NeuralPull loss. In addition, the idea of Gaussian normal constraints comes back from the era of Surface Splatting[1], and the idea of constraining gradients of SDF is also not new since NeuralPull[2] came out, methods like DiGS[3] and Towards Normal Consistency[4] also followed this idea. Thus, manipulating gradients and Gaussian normals has become a commonly used strategy, which does not originally come from NeuSG. Our reference to NeuSG in Line 175 is just for the flatten of Gaussians.
> > >
> > > Since we focus on Gaussians on the zero-level set, the implementations of our constraints on Gaussian normals and SDF gradient are much different from NeuSG. For example, we need to pull Gaussians onto the zero-level set, calculate SDF gradients at the pulled Gaussians on the zero-level set, and align Gaussian normal to these gradients (our Tangent loss in Eq.4 of the original paper). Moreover, we also pursue alignment across different level-sets by aligning gradients at queries (may appear on any level sets in the field) to the normal of Gaussians on the zero-level set (our orthogonal loss in Eq.6 of the original paper), which makes the gradients on other level sets orthogonal to the corresponding Gaussian disk on the zero-level set.
> > >
> > > We will make sure to include these discussions and NeuSG in our revision.
> > >
> > > **2. We are also more efficient than NeuSG**
> > >
> > > Since NeuSG is based on NeuS, which needs to sample points along a ray for integration, it inherits the drawback of NeuS, i.e., poor efficiency. On a scene, we just need about half an hour, while NeuSG needs about 16 hours. Obviously, our solution can better leverage the advantages of 3DGS than NeuSG.
> > >
> > > We will make sure to include these discussions and NeuSG in our revision.
> > >
> > > **3. Normal loss**
> > >
> > > Using either cosine or L1 to constrain normal is also widely used in point cloud normal estimation. As we stated in our response “7. Normal Loss” to reviewer SW8k, we did not see any differences on the performance for these two constraints on normal. We will include this ablation study in our revision.
> > >
> > > **4. 2DGS results**
> > >
> > > There is a numerical error in the mean operation. According to Table 2 in 2DGS paper, if we average the per-scene result provided in the table, we can get (0.36+0.23+0.13+0.44+0.16+0.26)/6=0.26, but 2DGS reported the mean value as 0.30 instead of 0.26.
> > >
> > >
> > > [1]. Zwicker M, Pfister H, Van Baar J, et al. Surface Splatting. Proceedings of the 28th annual conference on Computer graphics and interactive techniques. 2001: 371-378.
> > >
> > > [2]. Ma B, Han Z, Liu Y S, et al. Neural-Pull: Learning Signed Distance Function from Point clouds by Learning to Pull Space onto Surface. International Conference on Machine Learning. PMLR, 2021: 7246-7257.
> > >
> > > [3]. Ben-Shabat Y, Koneputugodage C H, Gould S. DiGS: Divergence guided shape implicit neural representation for unoriented point clouds. Proceedings of the IEEE/CVF Conference on Computer Vision and Pattern Recognition. 2022: 19323-19332.
> > >
> > > [4]. Ma B, Zhou J, Liu Y S, et al. Towards better gradient consistency for neural signed distance functions via level set alignment. Proceedings of the IEEE/CVF Conference on Computer Vision and Pattern Recognition. 2023: 17724-17734.

---

> > > > ### Comment · Reviewer_CCgx · 2024-08-12
> > > >
> > > > **Discussion about NeuSG**: While I agree that NeuSG optimizing a neural radiance field (specifically NeuS) makes it different from yours (where it is pure Gaussian splatting), I still think that they are related enough to warrant more discussion in the paper. I definitely agree that the extension to neural pull is novel and interesting, and overall think the proposed method is quite elegant (especially in relation to NeuSG). And of course the time performance in relation to NeuSG is amazing. I am glad the authors are willing to add more clarifications in the paper.
> > > >
> > > > **Normal loss**: glad you are happy to add this in the paper/supplement.
> > > >
> > > > **2DGS results**: It seems I was looking at v2 of their paper on arXiv (which came out after the submission deadline for NeurIPS), where their result on on TnT is a mean of 0.32 and their mean calculation seems fine. I see now that the mean calculation is wrong in v1 of their paper.
> > > >
> > > > **Overall**: The authors have addressed all my concerns and am glad for their efforts in the discussion period. I hope they add the clarifications and ablations that they promised as well as release their code. I wish to maintain my previous recommendation of accept (7).

---

> > > > > ### Author Response · Authors · 2024-08-12
> > > > > **Thanks for your accept recommendation**
> > > > >
> > > > > Dear reviewer CCgx,
> > > > >
> > > > > Thanks for your comments and valuable advice. We really appreciate your accept recommendation.
> > > > >
> > > > > Best,
> > > > >
> > > > > The authors

---

### Author Rebuttal · Authors · 2024-08-07

We thank reviewers for comments and highlighting our ***simple and interesting idea*** (Reviewer CCgx, 5rKf, ZhFP), ***good performance and visualization*** (Reviewer CCgx, ZhFP), ***well-written manuscript*** (Reviewer 5rKf). We have provided detailed responses to each reviewer's comments. All the figures and tables referenced are included in the rebuttal PDF. We gratefully thank the reviewers for taking the time to review our paper and for their valuable suggestions. We are looking forward to the feedback.

---

### Author Response · Authors · 2024-08-08
**We will be happy to take questions**

Dear reviewers,

We appreciate your comments and expertise. Please let us know if there is anything we can clarify further. We would be happy to take this opportunity to discuss with you.

Thanks,
The authors

---

### Comment · Area_Chair_DNXt · 2024-08-10

Hi reviewers,

Thank you for your hard work in reviewing the paper!
Please check out the authors' responses and ask any questions you have to help clarify things by Aug 13.

--AC

---

> ### Author Response · Authors · 2024-08-11
>
> Dear AC,
>
> Thanks for initiating the discussion. We are also looking forward to having a discussion with reviewers. It would be great if reviewers could let us know if their concerns are addressed after reading our rebuttal materials. We appreciate that a lot.
>
> Best,
>
> The authors

---

> > ### Author Response · Authors · 2024-08-13
> >
> > Dear AC,
> >
> > We are writing to report an issue of openreview today. It seems that it does not send emails to reviewers and authors when we submit our comments. We tried many times, and all of our authors did not receive emails like before. We are afraid that the reviewers may miss our comments, since the discussion period will end in one day.
> >
> > If you can see this message by any chance, can you inform reviewer ZhFP this issue and ask reviewer ZhFP to check our latest reply using personal emails?
> >
> > Really appreciate your help.
> >
> > Thanks,
> >
> > The authors

---

### Decision · Program_Chairs · 2024-09-25

**Decision:**

Accept (poster)

**Comment:**

This paper is a clear accept. It receives 2x accepts and 2x borderline accepts. The reviewers agreed that the proposed method is simple but effective with well-thought losses, the proposed method demonstrated state-of-the-art reconstruction results on the DTU and TNT datasets, with each component of the approach well-ablated. This work proposes an interesting method of combining a differentiable pulling operation with Gaussian Splatting for surface reconstruction to learn neural SDF for GS. The paper is also well-written and easy to follow. Although Reviewer SW8k initially raised several issues on lack of explanation, unsatisfactory results on TNT, inconsistency of quantitative results, lack of citation, etc, these concerns are well-addressed by the authors. The reviewer finally raised the score to borderline accept.